# TexQ: Zero-shot Network Quantization with Texture Feature Distribution Calibration

**Xinrui Chen, Yizhi Wang, Renao Yan*, Yiqin Liu, Tian Guan*, Yonghong He**
Tsinghua Shenzhen International Graduate School, Tsinghua University
{cxr22, yz-wang22, yra21, liuyiqin20}@mails.tsinghua.edu.cn
{guantian, heyh}@sz.tsinghua.edu.cn

## Abstract

Quantization is an effective way to compress neural networks. By reducing the bit width of the parameters, the processing efficiency of neural network models at edge devices can be notably improved. Most conventional quantization methods utilize real datasets to optimize quantization parameters and fine-tune. Due to the inevitable privacy and security issues of real samples, the existing real-data-driven methods are no longer applicable. Thus, a natural method is to introduce synthetic samples for zero-shot quantization (ZSQ). However, the conventional synthetic samples fail to retain the detailed texture feature distributions, which severely limits the knowledge transfer and performance of the quantized model. In this paper, a novel ZSQ method, TexQ is proposed to address this issue. We first synthesize a calibration image and extract its calibration center for each class with a texture feature energy distribution calibration method. Then, the calibration centers are used to guide the generator to synthesize samples. Finally, the mixup knowledge distillation module is introduced to diversify the synthetic samples for fine-tuning. Extensive experiments on CIFAR10/100 and ImageNet show that TexQ is observed to perform state-of-the-art in low bit width quantization. For example, when ResNet-18 is quantized to 3-bit, TexQ achieves a 12.18% top-1 accuracy increase on ImageNet compared to state-of-the-art methods.

## 1 Introduction

Limited by the computing capability of edge devices such as mobile phones, deep neural network models inevitably require compression for terminal scenarios [1, 2]. Compared to popular compressing methods such as pruning and distillation, quantization is easier to achieve lightweight and hardware compatibility [3, 4, 5, 6]. By compressing floating-point parameters to low-bit fixed-point integer, quantization reduces the resource footprint and improves computing efficiency. For example, when moving the parameters tensor (weights and activation) from 32 to 4 bits, the memory consumption is reduced by a factor of 8, while the computational overhead of matrix multiplication goes down by a factor of 64 at a squared rate. Most research on quantization focuses on quantization-aware training (QAT) [7, 8, 9, 10] and post-training quantization (PTQ) [11, 12, 13, 14, 15, 16]. QAT introduces fake quantization nodes when training and achieves performance close to full precision. However, it relies on the full training set and lacks a uniform specification, making it unfriendly to deploy. Thus, PTQ has gained attention, which only requires a small part of the training set to statistically optimize the quantization parameters [17, 18, 19, 20].

Due to privacy and security constraints, access to certain real data might be prohibited [21], such as patient medical images, confidential business information, etc. Therefore, zero-shot quantization

---

*Corresponding author

37th Conference on Neural Information Processing Systems (NeurIPS 2023).

(ZSQ) [22, 23, 21, 24, 25] is proposed to circumvent this limitations. Among previous studies, synthetic sample-based methods [21, 24, 25] have attracted attention by their excellent performance. Most studies synthesize samples that resemble the distribution of the real samples by extracting statistics in the full precision model, such as batch-normalization statistics [21], categorical labels [24], intermediate features [25], etc. However, despite a good fit to the model statistics and high classification confidence, there is still a huge gap between the texture feature distribution they contain and that of real samples. Such a phenomenon is likely to result in bad performance, as studies show that CNNs rely on texture feature to make decisions, for texture feature is "easy to learn" for CNNs [26, 27, 28]. In addition, the conventional multiple-constraint paradigm limits the diversity of synthetic samples [29, 30].

To tackle the above problems, the following exploratory experiments were carried out. Firstly, local binary pattern (LBP) [32] was introduced to characterize the texture feature. A batch of real samples and conventional synthetic samples with class confidence above 99.9% were extracted their LBP feature for clustering. As presented in Figure 1, the domain gap (point distribution) and diversity gap (point density) can be observed. We further explored the texture bias of the quantized model to prove the importance of texture feature. As shown in Figure 2, same model was respectively quantized with real samples and conventional synthetic samples, and tested in four image patterns: original, greyscale (removes color feature), binary (removes color and texture feature), and edge (retains shape feature and restores a few texture feature). It illustrates that removing texture feature causes a sharp performance decrease. The model quantized with synthetic samples drops off a lot on performance while restoring some of the texture feature in Figure 2d can recover its performance. These experiments imply that the ZSQ model is strongly biased towards

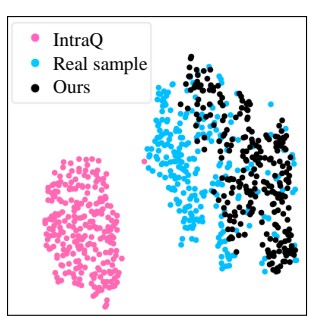

Figure 1: LBP feature clustering of samples from CIFAR10, IntraQ and our method. Visualization with t-SNE [31]

texture feature and tends to make decisions with it. Inspired by the experiments, an intuitive idea is to calibrate the synthetic samples to retain the texture feature distribution, which facilitates the model to learn and improve its performance.

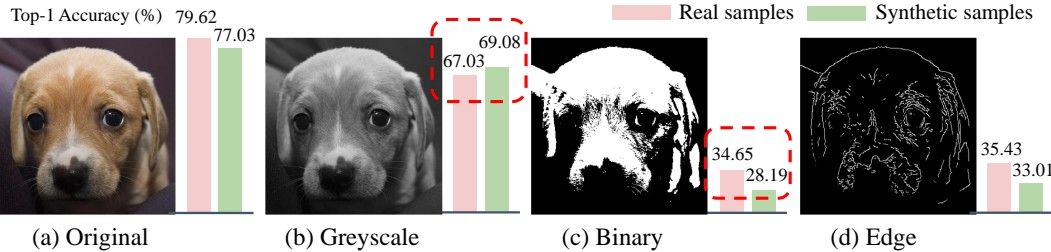

(a) Original     (b) Greyscale     (c) Binary     (d) Edge

Figure 2: Test results of real/synthetic samples quantized model on CIFAR10 in 4 image patterns.

In this work, we propose TexQ, a zero-shot quantization method using synthetic calibration centers to calibrate samples and retain texture feature distributions. We first generate a calibration images and its center for each class. Further, the centers guide the generator to synthesize samples. The two-stage synthetic sample paradigm alleviates the homogeneity caused by multiple constraints. Finally, mixup knowledge distillation is introduced to enhance sample diversity and avoid overfitting. With the above methods, the proposed TexQ performs state-of-the-art on various datasets and model settings.

## 2 Related work

### 2.1 Texture feature extraction

Texture feature is powerful visual cue that provides useful information in identifying objects or regions of interest in images [33]. To make use of these cues, texture feature extraction methods

has been widely used in natural image processing, among which filtering approaches are proven to be successful, including LAWS filters, dyadic Gabor filter banks, wavelet transforms and so on [34]. Studies [35, 36, 37] have shown that image texture feature are conductive to image classification and are class separable. For instance, [38] introduce GLCM and Gabor texture feature from regions of interest for better mammogram classification. Recently, many studies [39, 40, 41] identify the importance of textures feature for CNNs. [39] found that texture representations could capture the statistical characteristics of images for CNNs. [40] showed that classic CNNs were unable to recognize sketches where textures are missing and shapes are left. Similarly, [41, 42] validated that CNNs were biased towards textures than shapes, for example, ResNet-50 biased 77.9% texture.

## 2.2 Zero-shot quantization

Most zero-shot quantization (also called data-free quantization) studies try to recover the quantization error from three perspectives. The first perspective designs quantization parameters by using model properties [22, 23, 43, 44] without acquiring any data. For instance, Nagel et al. [22] proposed a scale-equivariance property of activation functions to equalize the weight ranges of the network. Meller et al. [44] highlighted an inversely proportional factorization of convolutional neural networks to decrease the degradation caused by quantization. The above methods avoid the access to data but suffer from severe performance degradation in low bit widths. For example, DFQ achieved a 0.11% top-1 accuracy in the 4-bit MobileNetV2 case [22, 24]. Therefore, more ZSQ methods resort to synthetic samples. The second perspective adopts optimization-based methods to synthesize samples by aligning the statistics in the full-precision network [21, 30, 29]. ZeroQ [21] adopted batch normalization statistics alignment to optimize the standard Gaussian noise. DSG [30] proposed a slack distribution alignment to diversify samples. IntraQ [29] highlighted the intra-class heterogeneity and retained this property in the synthetic samples for better performance. The third perspective adopts a generator to synthesize samples [24, 45, 46, 47, 48]. For example, GDFQ [24] proposed a knowledge-matching generator to produce synthetic data with labels by introducing the cross-entropy loss. ClusterQ [45] utilized the feature distribution alignment to imitate the distribution of real data. AdaDFQ [47] proposed a zero-sum game to adaptively regulate the adaptability of synthetic samples. The existing studies provide us effective method to fit statistics in full-precision models, however, none of these considered texture feature distribution in synthetic samples, though it is fundamental for CNNs to learn and make decisions. To the extent of our knowledge, this work is the first to consider texture feature in ZSQ, introducing a synthetic calibration center to calibrate synthetic samples.

# 3 Methodology

## 3.1 Preliminaries

### 3.1.1 Quantizer

Following ZeroQ [21], asymmetric quantization is adopted. Given a floating-point value $x_f$ (weights or activations) and quantization bit width $BW$, the quantized integer $x_q$ can be obtained as:

$$x_q = Clip(\lfloor x_f \cdot S - ZP \rceil, 0, 2^{BW} - 1) \tag{1}$$

where $S = \frac{2^{BW}-1}{x_{f\max}-x_{f\min}}$ is the scaling factor mapping the floating-point number to a fixed-point integer, $ZP = x_{f\min} \cdot S$ is the zero point mapping the floating-point minimum to zero, $\lfloor input \rceil$ rounds its input to the nearest integer. $Clip(tensor, r_{min}, r_{max})$ clamps the tensor elements to be between $r_{min}$ and $r_{max}$. The dequantized value $x_d$ can be obtained as:

$$x_d = \frac{x_q + ZP}{S}. \tag{2}$$

### 3.1.2 Data synthesis

PTQ requires a small real dataset $D = \{(x,\ y)\}$ containing samples $x$ and labels $y$. Similarly, most ZSQ introduce synthetic sample sets $\overline{D} = \{(\overline{x},\ \overline{y})\}$. To fit the batch normalization statistics (BNS) in the full-precision model ($F$), a basic principle [21, 45, 29] is using BNS alignment loss in Eq. 3.

$$\mathcal{L}_{BNS} = \sum_{l=1}^{L} \left\| \mu_l(\overline{x}) - \mu_l{}^F \right\|_2 + \left\| \sigma_l(\overline{x}) - \sigma_l{}^F \right\|_2, \tag{3}$$

where $\mu_l{}^F$ and $\sigma_l{}^F$ are the running mean and variance stored in the $l$-th BN layer of $F$. The mean and variance of the synthetic sample batch $\overline{x}$ in the $l$-th layer of $F$ are given by $\mu_l(\overline{x})$ and $\sigma_l(\overline{x})$.

Given generator $G$, standard Gaussian noise $z$, and a target label $\overline{y}$, the synthetic sample can be obtained as $\overline{x} = G\left(z|\overline{y}\right)$. The cross-entropy loss in Eq. 4 is used to generate label-oriented samples [24, 19].

$$\mathcal{L}_{CE} = \mathbb{E}_{(x,y)\sim\{(\bar{x},\bar{y})\}} \left[ Cross\text{-}entropy\left(F(\bar{x}), \bar{y}\right) \right]. \tag{4}$$

### 3.2 Texture feature distribution calibration

In this section, the calibration method and mixup knowledge distillation module are proposed. The framework of our TexQ is illustrated in Figure 3.

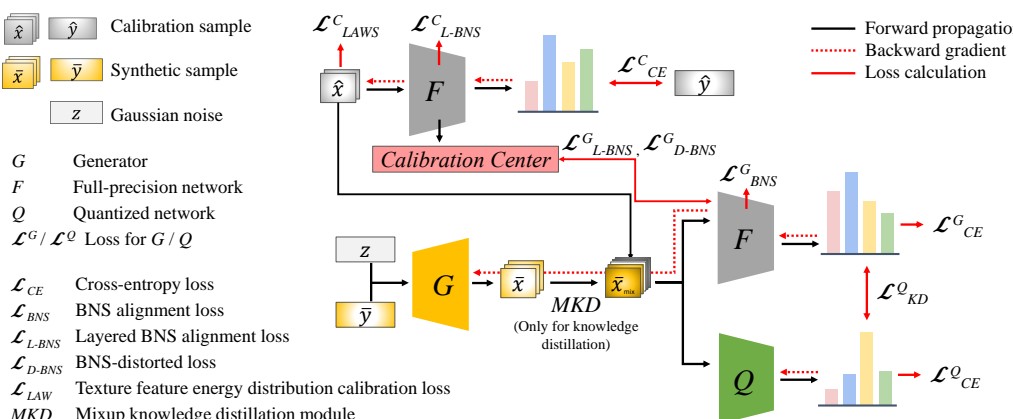

Figure 3: An overview of the proposed framework TexQ. Calibration centers are produced to guide $G$ to synthesize samples. $Q$ is fine-tuned with both calibration samples and synthetic samples.

### 3.2.1 Texture feature energy distribution calibration loss

To quantitatively measure the texture feature distribution, LAWS texture feature energy [49, 50] is introduced. We select four 1-dimensional filters provided by LAWS: $E_5$=[-1, -2, 0, 2, 1], $S_5$=[-1, 0, 2, 0, -1], $W_5$=[-1, 2, 0, -2, 1], $R_5$=[1, -4, 6, -4, 1], which stand for edge, spot, wave, and ripple. By convolving two 1-dimensional LAWS filters of size $1\times5$ with each other, a total of 16 2-dimensional filters of size $5\times5$ can be obtained. Each filter extracts a basic element of the texture, for example, $R_5R_5$ denotes a high-frequency point filter, and $E_5S_5$ denotes a V shape filter. To be specific, the calculation procedure of $R_5R_5$ is given as an example in Eq. 5.

$$R_5R_5 = R_5{}^T * R_5 = \begin{bmatrix} 1 \\ -4 \\ 6 \\ -4 \\ 1 \end{bmatrix} * \begin{bmatrix} 1 & -4 & 6 & -4 & 1 \end{bmatrix} = \begin{bmatrix} 1 & -4 & 6 & -4 & 1 \\ -4 & 16 & -24 & 16 & -4 \\ 6 & -24 & 36 & -24 & 6 \\ -4 & 16 & -24 & 16 & -4 \\ 1 & -4 & 6 & -4 & 1 \end{bmatrix} \tag{5}$$

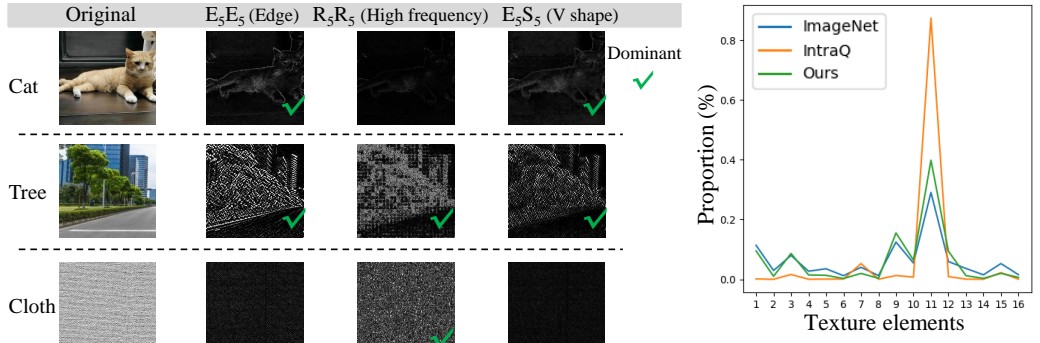

Figure 4: Features extracted with LAWS filters.

Figure 5: Visualization of distribution $T$ from different samples.

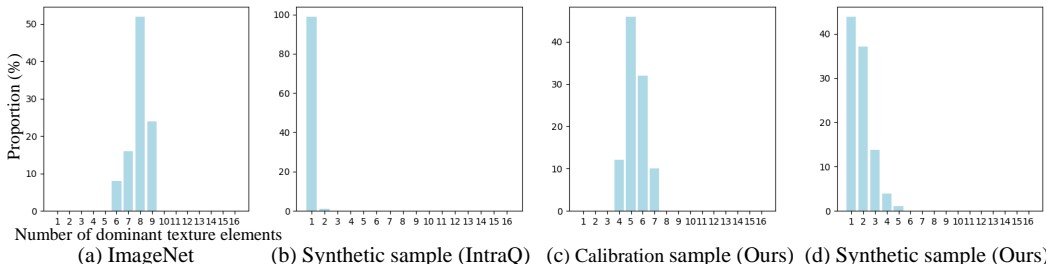

(a) ImageNet     (b) Synthetic sample (IntraQ)     (c) Calibration sample (Ours)     (d) Synthetic sample (Ours)

Figure 6: Dominant texture elements number distribution of samples from the ImageNet training set, IntraQ synthetic samples, calibration images as well as synthetic samples generated by our method. Synthetic samples were obtained with pre-trained ResNet-18.

Straightforward visualization of the texture feature extracted by LAWS texture feature filters is provided in Figure 4. It can be observed that different types of images contain different dominant texture feature. For example, the trees with lots of leaves including V-shape, high-frequency points and edge features, and the cat lying on the chair including mainly V-shape and edge features.

It is reasonable to evaluate texture distributions by analyzing basic elements, for complex texture feature are composed of simple elements. We apply 16 filters to obtain the texture feature energy distributions $T$ of a sample as shown in Figure 5. Assuming that the top-k elements in $T$ account for more than 80% are the dominant texture elements, the set of these k elements is recorded as $K$. As displayed in Figure 6, the real samples (Figure 6a) have at most 9 dominant texture elements, i.e., k $\leq 9$, while there is almost only 1 in the conventional synthetic samples (Figure 6b). To be detailed, in Figure 5, the distribution curve of the conventional synthetic sample presents a clear peak, indicating a single dominant element, which reduces the intra-class heterogeneity and the information contained. Unlike this, the curve of the real sample presents a detailed texture distribution. To retain this property, texture feature energy distribution calibration loss of Eq. 6 is proposed to dynamically adjust the detailed texture feature distribution of calibration images for each class.

$$\mathcal{L}^C{}_{LAWS} = \max\left(\|K_{\max} - \theta_U\|_2 - \varepsilon,\ 0\right) + \max\left(\|(1 - K_{\max}) - \theta_L\|_2 - \varepsilon,\ 0\right), \qquad (6)$$

where $K_{\max}$ is the largest element in $K$. The hyperparameters $\theta_U$, $\theta_L$, and $\varepsilon$ control the upper and lower bounds of the elements in $K$. Specifically, $\theta_U$ is the upper bound of $K_{\max}$, and $\theta_L$ is the lower bound of the sum of all elements in $K$ except $K_{\max}$. $\varepsilon$ is the softness factor to alleviate convergence problem. Both upper and lower bounds are used together to extract detailed distribution.

To introduce the calibration loss, a possible scheme is directly applying constraint to the generator. However, multiple constraints to generator results in slow iterations and homogeneous samples. We adopt a two-stage method, which includes an optimization method to synthesize accurate calibration centers for each class and a generation method to synthesize samples with the constraint on calibration centers.

### 3.2.2 Layered BNS alignment constraint

Considering that neural networks extract class-independent low-level features (e.g., texture and contours) in shallow layers and class-related high-level features in deep layers, the layered BNS alignment loss ($\mathcal{L}_{L\text{-}BNS}$) is proposed. We adopt loose constraints in shallow layers to facilitate the expression of texture feature and tightened constraints in deep layers to fit class-related information.

In calibration set generation stage, given the calibration set $C = \{(\widehat{x}, \widehat{y})\}$, the $\mathcal{L}^C_{L\text{-}BNS}$ of Eq. 7 constraints $C$ to align the BNS in $F$.

$$\mathcal{L}^C_{L\text{-}BNS} = \sum_{l=1}^{L} \left\| w_l \cdot (\mu_l(\widehat{x}_c) - \mu_l{}^F) \right\|_2 + \left\| w_l \cdot (\sigma_l(\widehat{x}_c) - \sigma_l{}^F) \right\|_2,$$
$$w_l = \begin{cases} 0.2, 1 \leq l < \left\lceil \frac{L}{2} \right\rceil - 2 \\ 1.1, \left\lceil \frac{L}{2} \right\rceil - 2 \leq l \leq L \end{cases},$$
(7)

where $L$ is the number of BN layers, $\mu_l(\widehat{x}_c)$ and $\sigma_l(\widehat{x}_c)$ denote the mean and variance of $C$ in the $l$-th layer of $F$. $w_l$ is the loss weight for the $l$-th layer. $\lceil input \rceil$ returns the smallest integer greater than its input.

In synthetic sample generation stage, $\mathcal{L}^G_{L\text{-}BNS}$ in Eq. 8 constrains the generator to synthesize samples that fit the calibration centers of corresponding class label $k$.

$$\mathcal{L}^G_{L\text{-}BNS} = \sum_{l=1}^{L} \left\| w_l \cdot \left[ \mu_l \left( \overline{x} | \overline{y} = k \right) - \mu_l{}^C \left( \widehat{x}_c | \widehat{y}_c = k \right) \right] \right\|_2$$
$$+ \left\| w_l \cdot \left[ \sigma_l \left( \overline{x} | \overline{y} = k \right) - \sigma_l{}^C \left( \widehat{x}_c | \widehat{y}_c = k \right) \right] \right\|_2, w_l = \begin{cases} 0, 1 \leq l < \left\lceil \frac{L}{2} \right\rceil - 2 \\ 1, \left\lceil \frac{L}{2} \right\rceil - 2 \leq l \leq L \end{cases},$$
(8)

where $\mu_l \left( \overline{x} | \overline{y} = k \right)$ and $\sigma_l \left( \overline{x} | \overline{y} = k \right)$ denote the mean and variance of the synthetic samples. $\mu_l{}^C \left( \widehat{x}_c | \widehat{y}_c = k \right)$ and $\sigma_l{}^C \left( \widehat{x}_c | \widehat{y}_c = k \right)$ are the corresponding mean and variance calibration center.

### 3.2.3 Mixup knowledge distillation module

Knowledge distillation is commonly used to transfer the output distribution from full-precision model $F$ to the quantized model $Q$ in the fine-tuning stage, which requires diversified samples. Some studies introduce Mixup [51, 48] augmentation to generate samples with mixed labels to fine-tune the quantized model with cross-entropy loss. However, the labels produced by the such methods are not accurate enough, and taking such mixed labels to fine-tune the quantized model tends to be disastrously misleading. Therefore, we advocate discarding mixed labels and taking only mixed samples to knowledge distillate the quantized model, and name this method mixup knowledge distillation. Specifically, as presented in Eq. 9, new sample $\overline{x}_{mix}$ is generated by weighted fusion of 2 samples $\overline{x}_i$ and $\overline{x}_j$ randomly selected in a batch for distillation. Mixup proportion $\lambda$ is sampled from a standard uniform distribution. The probability $p = 20\%$ is observed to perform best.

$$\overline{x}_{mix} = \begin{cases} \lambda \overline{x}_i + (1 - \lambda)\overline{x}_j, \; with \; probability \; p \\ \overline{x}_i, \; with \; probability \; 1 - p \end{cases}, \lambda \sim \mathrm{U}(0, 1)$$
(9)

## 3.3 Quantization process

### 3.3.1 Step1: Calibration set generation

In this stage, the calibration set with Gaussian noise initialization is optimized with the loss in Eq. 10, capturing the preferred texture of each class from $F$. We fix $F$ and back-propagate the loss to the calibration images. For each class, the calibration image is trained for at most 1500 iterations individually, with a warm-up of 150 iterations at half learning rate without $\mathcal{L}^C_{LAWS}$. Their BNS are extracted as the calibration centers to guide the generator in the next step.

$$\mathcal{L}^C = \mathcal{L}^C_{LAWS} + \alpha_1 \cdot \mathcal{L}^C_{L\text{-}BNS} + \alpha_2 \cdot \mathcal{L}^C_{CE}$$
(10)

### 3.3.2 Step2: Synthetic samples generation

In this stage, generator $G$ synthesize samples of different classes that fit corresponding calibration centers via $\mathcal{L}^G_{L\text{-}BNS}$. Following FDDA [19], BNS-distorted loss in Eq. 11 is introduced to avoid overfitting by Gaussian noise interfering with the calibration center. $\nu_\mu = 0.5$ and $\nu_\sigma = 1.0$ control the distortion degree of mean and variance.

$$
\begin{aligned}
\mathcal{L}^G_{D\text{-}BNS} = &\sum_{l=1}^{L} \left\| w_l \cdot \left[ \mu_l \left( \overline{x} | \overline{y} = k \right) - \mathcal{N} \left( \mu_l{}^C \left( \widehat{x_c} | \widehat{y_c} = k \right), \nu_\mu \right) \right] \right\|_2 \\
&+ \left\| w_l \cdot \left[ \sigma_l \left( \overline{x} | \overline{y} = k \right) - \mathcal{N} \left( \sigma_l{}^C \left( \widehat{x_c} | \widehat{y_c} = k \right), \nu_\sigma \right) \right] \right\|_2, w_l = \left\{ \begin{array}{l} 0, 1 \leq l < \lceil \frac{L}{2} \rceil - 2 \\ 1, \lceil \frac{L}{2} \rceil - 2 \leq l \leq L \end{array} \right. .
\end{aligned}
\tag{11}
$$

In addition, $\mathcal{L}^G_{CE}$ and $\mathcal{L}^G_{BNS}$ are used to align the label and BNS of $F$. Up to this point, the total loss employed by generator $G$ can be summarized in Eq. 12.

$$
\mathcal{L}^G = \mathcal{L}^G_{CE} + \alpha_3 \cdot \mathcal{L}^G_{BNS} + \alpha_4 \cdot \mathcal{L}^G_{L\text{-}BNS} + \alpha_5 \cdot \mathcal{L}^G_{D\text{-}BNS}.
\tag{12}
$$

### 3.3.3 Step3: Quantized model fine-tuning

To make full use of the data, we take both samples in $\overline{D}$ and $C$ as the input of $Q$ and apply cross-entropy loss $\mathcal{L}^Q_{CE}$ to fine-tune. Subsequently, both samples in $\overline{D}$ and $C$ are input into the mixup knowledge distillation module to produce mixup samples. With the input of mixup samples, Kullback-Leibler loss in Eq. 13 is applied to transfer the output of $F$ to $Q$.

$$
\mathcal{L}^Q_{KD} = \mathbb{E}_{(x,y) \sim C \cup \overline{D}} \left[ Kullback\text{-}Leibler \left( Q(\overline{x}_{mix}), F(\overline{x}_{mix}) \right) \right].
\tag{13}
$$

At this point, the overall loss for fine-tuning $Q$ can be summarized as:

$$
\mathcal{L}^Q = \mathcal{L}^Q_{CE} + \alpha_6 \cdot \mathcal{L}^Q_{KD}
\tag{14}
$$

## 4 Experiment

### 4.1 Experimental settings and details

We report top-1 accuracy on validation sets of CIFAR-10/100 [52] and ImageNet [53]. Networks selected include ResNet-20 [54] for CIFAR-10/100, ResNet-18, MobileNetV2 [55] and ResNet-50 for ImageNet. All experiments are implemented with Pytorch [56] via the code of FDDA [19] and IntraQ [29], and run on an NVIDIA GeForce RTX 3090 GPU. Calibration images are iterated with a constant learning rate of 0.05. Generator is imported from GDFQ [24] with a initial learning rate of 1e-3 multiplied by 0.1 every 100 epochs. In fine-tuning, batchsize is 128 for CIFAR-10/100 and 16 for ImageNet, adjusting by cosine annealing [57]. We warm up $G$ for 50 epochs, then update $G$ and $Q$ for 450 epochs. The optimal configurations on trade-off parameters from $\alpha_1$ to $\alpha_6$ obtained by grid search are 2, 10, 0.4, 0.02, 1.8, and 20.

### 4.2 Performance comparison

To demonstrate the efficacy of our TexQ, we conduct experiment in 3/4-bit case, since high accuracy can be easily achieved with a larger bit width. For instance, the advanced AdaDFQ [47] trails full precision with a 0.08% top-1 accuracy in 5-bit ResNet-20 case. In this section, WBAB indicates the weights and activations are quantized to B-bit. Best results in boldface.

### 4.2.1 CIFAR-10/100

We compare the performance against the advanced ZSQ methods on CIFAR-10/100. As presented in Table 1, our TexQ is observed to achieve state-of-the-art among the competitors, improving the top-1 accuracy by 3.13%/1.58% in 3-bit CIFAR-10/100 case compared to the advanced AdaDFQ. Similar

results can be observed in the 4-bit case. In particular, the top-1 accuracy of our TexQ exceeds that of the same framework with real data in 4-bit case, demonstrating that the proposed TexQ can fully extract the feature distribution in simple datasets.

Table 1: Results of ResNet-20 on CIFAR-10/100.

| Bit width | Method | Top-1 Accuracy(%) | |
|---|---|---|---|
| | | CIFAR-10 | CIFAR-100 |
| | Full-precision | 93.89 | 70.33 |
| | Real data | 91.52 | 66.80 |
| W4A4 | GDFQ [24] (ECCV 2020) | 90.11 | 63.75 |
| | ARC [58] (IJCAI 2021) | 88.55 | 62.76 |
| | Qimera [48] (NeurIPS 2021) | 91.26 | 65.10 |
| | IntraQ [29] (CVPR 2022) | 91.49 | 64.98 |
| | ARC+AIT [59] (CVPR 2022) | 90.49 | 61.05 |
| | AdaSG [46] (AAAI 2023) | 92.10 | 66.42 |
| | AdaDFQ [47] (CVPR 2023) | 92.31 | 66.81 |
| | **TexQ (Ours)** | **92.68** | **67.18** |
| W3A3 | GDFQ [24, 47] (ECCV 2020) | 75.11 | 47.61 |
| | ARC [58] (IJCAI 2021) | - | 40.15 |
| | Qimera [48, 47] (NeurIPS 2021) | 74.43 | 46.13 |
| | IntraQ [29] (CVPR 2022) | 77.07 | 48.25 |
| | ARC+AIT [59] (CVPR 2022) | - | 41.34 |
| | AdaSG [46] (AAAI 2023) | 84.14 | 52.76 |
| | AdaDFQ [47] (CVPR 2023) | 84.89 | 52.74 |
| | **TexQ (Ours)** | **86.47** | **55.87** |

### 4.2.2 ImageNet

We further compare with competitors on challenging ImageNet, the results are presented in Table 2.

Table 2: Results of ResNet-18, MobileNetV2 and ResNet-50 on ImageNet.

| Bit width | Method | Top-1 Accuracy(%) | | |
|---|---|---|---|---|
| | | ResNet-18 | MobileNetV2 | ResNet-50 |
| | Full-precision | 71.47 | 72.49 | 77.73 |
| W4A4 | GDFQ [24] (ECCV 2020) | 60.60 | 59.43 | 54.16 |
| | ZAQ [25] (CVPR 2021) | 52.64 | 0.10 | 53.02 |
| | ARC [58] (IJCAI 2021) | 61.32 | 60.13 | 64.37 |
| | Qimera [48] (NeurIPS 2021) | 63.84 | 61.62 | 66.25 |
| | IntraQ [29] (CVPR 2022) | 66.47 | 65.10 | - |
| | ARC+AIT [59] (CVPR 2022) | 65.73 | 66.47 | 68.27 |
| | AdaSG [46] (AAAI 2023) | 66.50 | 65.15 | 68.58 |
| | AdaDFQ [47] (CVPR 2023) | 66.53 | 65.41 | 68.38 |
| | **TexQ (Ours)** | **67.73** | **67.07** | **70.72** |
| W3A3 | GDFQ [24, 47] (ECCV 2020) | 20.23 | 1.46 | 0.31 |
| | ARC [58] (IJCAI 2021) | 23.37 | 14.30 | 1.63 |
| | Qimera [48, 47] (NeurIPS 2021) | 1.17 | - | - |
| | AdaSG [46] (AAAI 2023) | 37.04 | 26.90 | 16.98 |
| | AdaDFQ [47] (CVPR 2023) | 38.10 | 28.99 | 17.63 |
| | **TexQ (Ours)** | **50.28** | **32.80** | **25.27** |

**Bit width** For low bit width case, conventional ZSQ methods suffer from a huge accuracy loss, while our TexQ performs good generalization capability. Specifically, for 4-bit ResNet-18 case,

TexQ achieving 67.73% accuracy, outperforming AdaDFQ by 1.2%. In 3-bit case, TexQ reaches 50.28% outstanding accuracy, outperforming AdaDFQ by 12.18%, with a standard deviation of 0.34 in 5 repeated experiments. Similar results were obtained in MobileNetV2.

**Network Size**  Larger models are likely lead to poor performance with existing ZSQ methods, especially in low bit width case. For example, in 3-bit case, Qimera [48, 47] performs well on the small ResNet-20, while achieves only 1.17% accuracy on the larger ResNet-18. In contrast, our TexQ achieved 55.87% ultra-low when quantizing ResNet-18 to 3-bit, and leads AdaDFQ with 7.64% on 3-bit ResNet50 case. These results again demonstrate the effectiveness of our method.

### 4.3  Ablation study

**Hyperparameters**  After empirical initialization, the hyperparameters k, $\theta_U$, $\theta_L$, and $\varepsilon$ are searched for their optimal value with grid search by quantizing ResNet-18 to 3-bit on ImageNet, as displayed in Figure 7. The optimal configurations are k=9, $\theta_U$=0.3, $\theta_L$=0.5, and $\varepsilon$=0.015. The hyperparameter k represents the number of dominant texture elements, whose optimal value of k=9 is consistent with what we observe in the real samples. The sum of the optimal values of $\theta_U$ and $\theta_L$ is 0.8, which fits our assumptions about the proportion of dominant texture elements. To avoid complex searches, these parameters were used for all experiments. While this may not be optimal for all networks, it is sufficient to exhibit the advanced performance of TexQ.

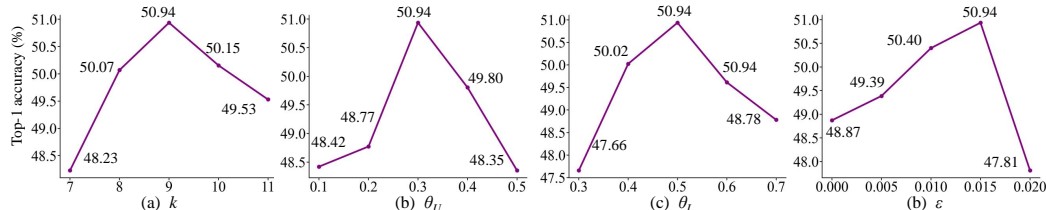

Figure 7: Influence of the hyperparameters.

**Modules**  Ablation on key modules including $\mathcal{L}^C{}_{LAWS}$ (Eq. 6), $\mathcal{L}^C{}_{L\text{-}BNS}$ (Eq. 7) and mixup knowledge distillation (MKD, Eq. 9) is conducted. As presented in Table 3, dropping one or two of them results in an accuracy loss. The largest accuracy loss (8.37%) is observed when both $\mathcal{L}^C{}_{LAWS}$ and $\mathcal{L}^C{}_{L\text{-}BNS}$ are removed, which indicates their cooperative relationship: $\mathcal{L}^C{}_{L\text{-}BNS}$ loosens the shallow layers constraint, facilitating $\mathcal{L}^C{}_{LAWS}$ to calibrate.

Table 3: Ablations on modules. We report the top-1 accuracy of 3-bit ResNet-18 on ImageNet.

| $\mathcal{L}^C{}_{LAWS}$ | $\mathcal{L}^C{}_{L\text{-}BNS}$ | MKD | Top-1 Accuracy(%) |
|:---:|:---:|:---:|:---:|
| ✓ | ✓ | ✓ | **50.94** |
| ✓ | ✓ |   | 49.03 |
| ✓ |   | ✓ | 43.31 |
|   | ✓ | ✓ | 49.63 |
| ✓ |   |   | 42.65 |
|   | ✓ |   | 49.00 |
|   |   | ✓ | 42.57 |

## 5  Discussion

### 5.1  Why does TexQ work?

**Mitigating the domain gap**  Our synthetic samples with calibration match the real distribution better, as presented in Figure 1 and Figure 6. Further, Figure 5 shows our synthetic sample has a more detailed texture distribution similar to the real one.

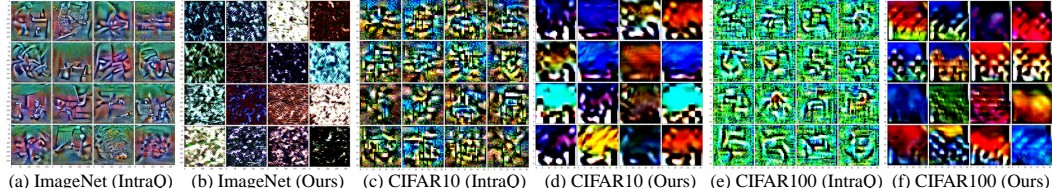

(a) ImageNet (IntraQ)   (b) ImageNet (Ours)   (c) CIFAR10 (IntraQ)   (d) CIFAR10 (Ours)   (e) CIFAR100 (IntraQ)   (f) CIFAR100 (Ours)

Figure 8: Visualization of synthetic samples. For ImageNet and CIFAR10/100, samples are obtained with ResNet-18 and ResNet-20, respectively. The advanced IntraQ was selected as a competitor.

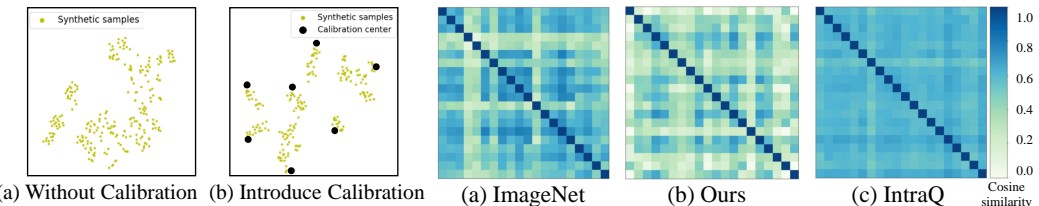

(a) Without Calibration  (b) Introduce Calibration     (a) ImageNet     (b) Ours     (c) IntraQ

Figure 9: Visualization of the running mean of BNS in ResNet-20.

Figure 10: Cosine similarity confusion matrix of samples. Synthetic samples are obtained with ResNet-18.

**Increasing inter-class distance**    As displayed in Figure 9a, the class centers of samples from conventional methods is unclear, limiting the inter-class distance. The situation is improved by introducing calibration centers, as shown in Figure 9b.

**Enhancing sample diversity**    Figure 1 shows our synthetic samples scatter a lot compared with IntraQ, implying the enhancement of sample diversity. To put it bluntly, synthetic images from different methods are visualized in Figure 8, showing the improvement in color and texture diversity of our samples. Further, the intra-class diversity is observed through cosine similarity confusion matrix of samples in Figure 10, showing that our samples reach a similar diversity to the real ones.

## 5.2   Limitations and future work

This work is built upon the concept of "texture", and the texture filters adopted are designed manually. Thus, the application to other tasks with different modality would be limited. We envision adding training process to the filters, which would generalize the calibration method proposed to other modality. Low-bit quantization remains a challenge. For instance, TexQ achieve 70.72% accuracy by quantizing ResNet-50 to 4-bit, while it drops to 25.27% in 3-bit case. The introduction of advanced distillation methods holds promise for more better results.

## 6   Conclusion

In this paper, we observe a non-negligible detailed texture distribution in the real samples. To retain this property in synthetic samples, we introduce synthetic calibration images and centers to calibrate the generator. To diversify the samples, mixup knowledge distillation module is introduce to create diversified samples for fine-tuning. Extensive experiments show our state-of-the-art performance on mainstream networks and datasets, especially for low bit width quantization.

## Acknowledgment

This work was supported by the Science and Technology Research Program of Shenzhen City (No. KCXFZ20201221173207022, No.WDZC2020200821141349001) and the Jilin Fuyuan Guan Food Group Co., Ltd.

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
