# OpenReview forum: "TexQ: Zero-shot Network Quantization with Texture Feature Distribution Calibration"
_NeurIPS.cc/2023/Conference — NeurIPS 2023 poster_

### Official Review · Reviewer_h8LC · 2023-06-27

**Soundness:** 3 good
**Presentation:** 2 fair
**Contribution:** 3 good
**Rating:** 6
**Confidence:** 5

**Summary:**

This paper starts from an interesting viewpoint that the synthetic of the previous ZSQ method usually fails to model the similar text feature like real data. As a result, the authors suggest retaining the texture feature. At first, they synthesize calibration images with a LAWS texture feature energy preserve loss. Then, the calibration images are used to provide mean and variance centers for each class, which is then used to guide the generator to synthesize samples. At last, they proposed using a mix-up strategy to further augment the synthetic data.

**Strengths:**

1, The viewpoint is interesting and persuasive. Also, the results do demonstrate the effectiveness of their method.
2, Sec 5.1 provide many meaningful discussion, which makes this paper more convincing.


**Weaknesses:**

1, To be honest, the writing of this paper is not good enough and I get lost in many parts.

For example, Line 171, ‘Q always faces linear decisions when inferencing’ is very strange. I can't understand the meaning behind it.

Line 166, where µ_l (\bar(x)| \bar(y) = k) and σ_l (\bar(x)| \bar(y) = k) should be the calibrating mean and variance center since calibration set C = {( \bar(x), \bar(y))} in Line 160.

2, Why do the authors want to use a calibration center in Step2: Synthetic samples generation? The paper lacks some discussion.

3, The principle behind Calibration set generation and Synthetic samples generation is not clear. Why do you still need Synthetic samples, even though you already have Calibration set. I know such a way could give a better performance. The authors should provide some ablation study when only using Calibration set.

4, The role of Eq6. The K set only selects top-k texture elements that account for more than 80%. However, the images are randomly initialized from Gaussian noise. I am wondering if the distribution of the K set keeps the same for each synthetic image. For example, R5R5 is very fit for Gaussian noise and is selected for each image.


**Questions:**

1, Line 193, what is \bar(D)?
2, It actually uses synthetic data from the generator and from direct optimization. It may an unfair comparison.


**Limitations:**

The authors do provide a discussion about the limitations.

---

> ### Author Rebuttal · Authors · 2023-08-08
>
> ### **We would like to thank the reviewer for the thorough and thoughtful revision of the paper. In the following, we will address each point raised.**
> We not only respond to or clarify the points but have also made changes accordingly to the main text to make them more clear to future readers.
>
> ---
>
> **Q1: Line 193, what is $\bar{D}$?**
>
> **A1:  $\bar{D}$ denotes the synthetic sample set  $\bar{D} =  \\{\( \bar{x},  \bar{y}\)\\} $. The definition is presented in Section 3.1.2 Data synthesis (line 117).**
>
> ---
>
> **Q2: It actually uses synthetic data from the generator and from direct optimization. It may an unfair comparison.**
>
> **A2: Zero-shot quantization (ZSQ) requires strictly avoiding real data. In our work, both the synthetic dataset $\bar{D}$ and the calibration dataset $C$ are synthesized, so it conforms to the fairness of ZSQ comparison.**
>
> ---
>
> **Q3: Line 171, ‘Q always faces linear decisions when inferencing’ is very strange. I can't understand the meaning behind it.**
>
> **A3: We rewrite this paragraph and hope it is clearer.**
>
> Line 171: Our motives lie in the fact that quantized model Q always faces samples with soft labels when inferencing, we hope to synthesize samples with soft labels, too.
>
> ---
>
> **Q4: Line 166, where $µ_l (\bar{x}| \bar{y} = k)$ and $σ_l (\bar{x}| \bar{y} = k)$ should be the calibrating mean and variance center since calibration set $C = \{\( \bar{x}, \bar{y}\)\}$ in Line 160.**
>
> **A4: We checked the expression and found no problem, but we would like to make this point more clear to future readers.**
>
> * In the synthetic sample generation stage, we used layered BNS alignment to constrain the generator to synthesize samples that fit the calibration centers. Therefore, Eq. 8 includes the calibrating center and BNS of synthetic samples. In other words, with this method, texture feature distribution knowledge is transferred from the calibrating center to the synthetic samples.
> * We will add superscript $^\bar{D}$ to the variables involved in the synthetic samples to clarify the data source.
>
> ---
>
> **Q5: Why do the authors want to use a calibration center in Step2: Synthetic samples generation? The paper lacks some discussion.**
>
> **A5: We will supplement the relevant details in Section 3.2.2 (line 155).**
>
> * The introduction of calibration centers is a strategy for information transfer, which is also done in FDDA[12] and ClusterQ[17].
> * The principle is that the batch normalized statistics (BNS) of the convolutional neural network are highly related to the class, so introducing a calibration center in Step2 means transferring the information of the calibration center extracted in Step 1 to the generator.
>
> ---
>
> **Q6: The principle behind Calibration set generation and Synthetic samples generation is not clear. Why do you still need Synthetic samples, even though you already have Calibration set. I know such a way could give a better performance. The authors should provide some ablation study when only using Calibration set.**
>
> **A6: These are commonly used information transfer strategies, and we will supplement the detailed motivation in Section 3.3.2 (line 185) for easy understanding by readers:**
>
> * Distillation (optimization) [14] and generation (generator) [13] are two commonly used methods for synthesizing data. Distillation scheme directly optimizes Gaussian noise to synthesize samples, which takes a long time but less information transfer links, so the knowledge is more accurate. Generation scheme require training a generator. It is suitable for mass data generation, but the increase in information transfer links is not conducive to accurate knowledge transfer.
>
> * Our work combines the best of both schemes. We use the optimization method to synthesize accurate calibration centers for each class and introduce calibration centers into the generator to synthesize samples in batches, taking both accuracy and speed into account.
>
> * In addition, another possible scheme is to directly apply texture feature constraints to the generator. However, applying calibration lost to generator results in slow iterations and homogeneous samples. So, this option was not adopted.
>
> We conducted ablation studies on calibration samples and synthetic samples. **The three sample synthesis schemes and their results are shown in Table G. The scheme adopted in this paper is the optimal solution.**
>
> Due to characters limit, ***Tabel G are displayed in the global response area.***
>
> ---
>
> **Q7: The role of Eq6. The K set only selects top-k texture elements that account for more than 80%. However, the images are randomly initialized from Gaussian noise. I am wondering if the distribution of the K set keeps the same for each synthetic image. For example, R5R5 is very fit for Gaussian noise and is selected for each image.**
>
> **A7: The K sets are similar when Gaussian noise is initialized, but finally they become different, especially for different classes of images.**
>
> * Many studies [8-10] have shown that texture features are conducive to classifying images, so the final K sets are various from different classes of images.
> * R5R5 is very suitable for Gaussian noise, so it is selected by most images in the initial stage, and acts as a noise reduction function.
>
> ```
> [8] Filtering for texture classification: A comparative study. IEEE Transactions on pattern analysis and machine intelligence. 1999.
> [9] Image Classification Using Laws' Texture Energy Measures. 1987.
> [10] Texture and shape biased two-stream networks for clothing classification and attribute recognition. CVPR. 2020.
> [12] Fine-grained data distribution alignment for post-training quantization. ECCV. 2022.
> [13] Generative low-bitwidth data free quantization. ECCV. 2020.
> [14] Intraq: Learning synthetic images with intra-class heterogeneity for zero-shot network quantization. CVPR. 2022.
> [17] ClusterQ: Semantic Feature Distribution Alignment for Data-Free Quantization. arXiv. 2022.
> ```

---

> > ### Comment · Reviewer_h8LC · 2023-08-18
> >
> > The author solves my problems and I am willing to raise my rating.

---

### Official Review · Reviewer_qhGS · 2023-07-01

**Soundness:** 3 good
**Presentation:** 3 good
**Contribution:** 3 good
**Rating:** 7
**Confidence:** 2

**Summary:**

TexQ is a novel zero-shot quantization (ZSQ) method that addresses the limitations of conventional synthetic samples in retaining texture feature distributions. It achieves state-of-the-art results in ultra-low bit width quantization, with a significant accuracy increase compared to existing methods on ImageNet.

**Strengths:**

Strengths are shown below:

1. This paper tried to exploit the important direction for inevitable privacy and security.
2. The performance on the benchmark datasets is promising.
3. The paper is well-organized.

**Weaknesses:**

1. Maybe the structures used are insufficient with ResNet-18 and MobileNet-V2. What about the most commonly used ResNet-50?
2. Maybe the datasets verified are insufficient with Cifar and ImageNet. What about the others tasks, such as CoCo of detection.

**Questions:**

The authors could refer to the Weaknesses. Additionally, I also have some open questions to discuss.

1. The authors could discuss how to ensure the generalization of the method since the calibration is conducted on a specific dataset.
2. Could the method be used in the Transformer structure?

---

> ### Author Rebuttal · Authors · 2023-08-09
>
> ### **Thank you for your positive review and constructive comments. We have performed some of them and will include them in the camera-ready version.**
>
> ---
>
> **Q1: Maybe the structures used are insufficient with ResNet-18 and MobileNet-V2. What about the most commonly used ResNet-50?**
>
> **A1: Thank you for your advice. We supplement the experiments on ResNet-50, and the results are shown in Table D.**
>
> * **We continue to show advanced performance on the ResNet50, such as we lead AdaDFQ with 7.64%/2.34% Top1 accuracy on 3/4 bits case**.
> * For our experiments, we follow the settings of Section 4.1 except that we raise the weight of L-BNS loss and BNS alignment loss to ensure the convergence of the model.
>
> ---
>
> #### **Table D. Top 1 accuracy (%) results of ResNet50 on ImageNet. WBAB indicates the weights and activations are quantized to B-bit.**
> |     Methods    |     W4A4    |     W3A3    |
> |---|---|---|
> |     GDFQ(ECCV   2020)    |     54.16    |     0.31    |
> |     ZAQ   (CVPR 2021)    |     53.02    |     -    |
> |     ARC   (IJCAI 2021)    |     64.37    |     1.63    |
> |     Qimera   (NeurIPS 2021)    |     66.25    |     -    |
> |     ARC+AIT   (CVPR 2022)    |     68.27    |     -    |
> |     AdaSG   (AAAI 2023)    |     68.58    |     16.98    |
> |     AdaDFQ   (CVPR 2023)    |     68.38    |     17.63    |
> |     **TexQ   (Ours)**    |     **70.72**    |     **25.27**    |
>
> ---
>
> **Q2: Maybe the datasets verified are insufficient with Cifar and ImageNet. What about the others tasks, such as CoCo of detection.**
>
> **A2: Generally, most zero-shot quantization works [12-16] in the research community are validated in CIFAR10/100 and ImageNet, for these datasets are persuasive and convincing.**
> * For our work, we are the first to propose the idea of texture calibration and validate it, which is the focus.
> * Expanding to downstream tasks will be a new research hotspot, which will also be the direction of our future exploration.
>
> ---
>
> **Q3: The authors could discuss how to ensure the generalization of the method since the calibration is conducted on a specific dataset.**
>
> **A3: This is an open question worth exploring.**
>
> * In our work, we apply the same texture feature distribution calibration method (including hyperparameters) across different image sizes and scales of datasets. Advanced results show that our method is capable of generalization, even if these settings are not optimal for every dataset.
> * Further considering a specific dataset, the following two aspects can be explored. First, trainable texture feature filters can be used to extract specific features on a specific dataset. Second, adaptive method can be introduced to balance multiple losses.
>
> ---
>
> **Q4: Could the method be used in the Transformer structure?**
>
> **A4: The calibration idea proposed can be transferred to other computer vision models including the transformer structure.**
> * However, currently CNN and Transformer require different quantizers, and the quantization framework we adopted does not support the transformer structure model.
> * More importantly, our main contribution is to propose the idea of texture calibration in zero-shot quantization and verify it. We will introduce the quantizer for the transformer structure in future work.
>
> ---
>
> ```
> [12] Zhong, Yunshan, et al. Fine-grained data distribution alignment for post-training quantization. European Conference on Computer Vision. Cham: Springer Nature Switzerland, 2022.
> [13] Xu, Shoukai, et al. Generative low-bitwidth data free quantization. Computer Vision–ECCV 2020: 16th European Conference, Glasgow, UK, August 23–28, 2020, Proceedings, Part XII 16. Springer International Publishing, 2020.
> [14] Zhong, Yunshan, et al. Intraq: Learning synthetic images with intra-class heterogeneity for zero-shot network quantization. Proceedings of the IEEE/CVF Conference on Computer Vision and Pattern Recognition. 2022.
> [15] Qian, Biao, et al. Rethinking data-free quantization as a zero-sum game. arXiv preprint arXiv:2302.09572. 2023.
> [16] Qian, Biao, et al. Adaptive Data-Free Quantization. Proceedings of the IEEE/CVF Conference on Computer Vision and Pattern Recognition. 2023.
> ```

---

> > ### Comment · Reviewer_qhGS · 2023-08-17
> >
> > Thank the authors for their explanation. All my concerns are addressed. I support accepting this paper from my perspective, so I decided to increase my score to 7.

---

### Official Review · Reviewer_F97y · 2023-07-02

**Soundness:** 3 good
**Presentation:** 3 good
**Contribution:** 2 fair
**Rating:** 5
**Confidence:** 4

**Summary:**

This work proposes TexQ, which targets keeping the texture information of the synthetic samples of zero-shot quantization. The texture feature energy distribution calibration method is applied to the synthesized samples, and mixup knowledge distillation is introduced to improve the diversity of the synthetic samples. Extensive experiments of ResNet, MobileNet on CIFAR, and ImageNet datasets prove the effectiveness of this method.

**Strengths:**

* The paper is well-organized and easy to follow. Fig.3 is an excellent visualization of the proposed system architecture.

* To measure the texture feature distribution for the input sample of quantization is novel. The comparison (Fig. 4/5) of synthetic samples and natural images is insightful and may help future research on the ZSQ.

* The experiment results are convincing and high accuracy (especially for 3-bit) demonstrates the effectiveness of TexQ.

**Weaknesses:**

* The introduction of the concept "LAWS texture feature energy" needs to be improved. The example given in Eq.5 is not straightforward. Visualization of features extracted using $E_5, S_5, W_5, R_5$ would be better. Why these "texture" feature is important to retain should also be included more in the discussion.
* There are too many artifact weighting coefficients $\alpha_i, i \in [1,5]$ to balance different loss function. The author claims in Sec. 4.1 that they "empirically" select the value. An ablation study or sensitivity analysis on these parameters would be better to prove that the proposed loss is solid and that better performance does not come from a grid search of the parameters.
* The method is only validated on CNNs. The experiment results on Vision Transformer would be a plus.

**Questions:**

* Most previous work uses the term ''data-free quantization (DFQ)'' instead of "zero-shot quantization (ZSQ)." I want the author to make sure that these two terminologies are identical.
* What is the actual (visual/physical/signal) meaning of the feature extracted using the filters $E_5, S_5, W_5, R_5$?
* Can TexQ be transferred to language tasks without "texture" in the data?


**Limitations:**

The author includes a discussion of the problems to be solved in Section 5.2, which is great. One crucial limitation I would like to raise is that this work is built upon the concept of "texture" and is **only meaningful for visual samples**. The application to other tasks with different modality (language, speech, etc.) would be limited.

---

> ### Author Rebuttal · Authors · 2023-08-09
>
> ### **Thank you for providing new ideas for straightforward visualization and interesting transferred language tasks. We would like to address the concerns below:**
>
> ---
>
> **Q1: The introduction of the concept "LAWS texture feature energy" needs to be improved. The example given in Eq.5 is not straightforward. Visualization of features extracted using $E_5, S_5, W_5, R_5$ would be better. Why these "texture" feature is important to retain should also be included more in the discussion. What is the actual (visual/physical/signal) meaning of the feature extracted using the filters?**
>
> **A1: We agree to provide more details to introduce the concept of "LAWS texture feature energy"**
>
> * More background in the Introduction and Methods (Section 3.2.1) will be provided to introduce the concept of "LAWS texture feature energy", including the details of LAWS texture feature filters [8] and their application [9-10]. This helps future readers to understand our work.
>
> * Visualization is a good idea. Straightforward visualization of the texture features extracted will be provided in Figure 5 (line 127) after introducing the concept of "LAWS texture feature energy".
>
> * In addition, we will add the actual signal meaning and function of the filters in Section 3.2.1 (line 126). $E_5, S_5, W_5, R_5$ are series of texture feature filters mnemonics standing for Edge, Spot, Wave, and Ripple. The filtered feature map reflects the degree of matching between the texture and the filters. Complex texture feature distribution is composed of simple texture elements, so it is reasonable to introduce basic texture filters to characterize the texture feature distribution of images. These discussions will be supplemented in the camera-ready version.
>
> ---
>
> **Q2: There are too many artifact weighting coefficients to balance different loss function. The author claims in Sec. 4.1 that they "empirically" select the value. An ablation study or sensitivity analysis on these parameters would be better to prove that the proposed loss is solid and that better performance does not come from a grid search of the parameters.**
>
> **A2: Thank you for this suggestion. We agree that the adopted hyperparameter settings are not globally optimal. We adopted the settings of relevant studies [12-14] to avoid time costly grid search.**
> * More importantly, we focus on verifying the texture feature distribution calibration method, and empirical settings are conducive to fair comparison with the same type of work.
> * Experiments show that our idea works even when we configure hyperparameters empirically, which is enough to verify our idea.
>
> ---
>
> **Q3: Can TexQ be transferred to language tasks without "texture" in the data?**
>
> **A3: That's an interesting question.**
>
> * In computer vision, texture is an important feature of images. Is there any similar feature in the field of language such as speech? I think the answer is absolutely yes.
>
> * I agree with the constructive comment made by reviewer yCFw, that we can further introduce trainable filters to achieve automatic feature extraction. With such methods, our method holds promise for applications in other fields.
>
> ---
>
> **Q4: The method is only validated on CNNs. The experiment results on Vision Transformer would be a plus.**
>
> **A4: Thank you for this suggestion, the ideas of our work can be transferred to other visual models.**
>
> * Unfortunately, the quantizer adopted in this paper does not involve the quantizer of ViT.
> * The emphasis of this paper is to put forward the idea of texture calibration and verify it. We take Vision Transformer into consideration in the subsequent work.
>
> ---
>
> **Q5: Most previous work uses the term ''data-free quantization (DFQ)'' instead of "zero-shot quantization (ZSQ)." I want the author to make sure that these two terminologies are identical.**
>
> **A5: The terminologies ''data-free quantization (DFQ)' and 'zero-shot quantization (ZSQ)' are identical in the research community.** We will illustrate these terms in Related work (line 86) and modify the expression in the camera-ready version.
>
> ---
>
> ```
> [8] Randen, Trygve, and John Hakon Husoy. Filtering for texture classification: A comparative study. IEEE Transactions on pattern analysis and machine intelligence. 1999.
> [9] Gillett, Will D. Image Classification Using Laws' Texture Energy Measures. 1987.
> [10] Zhang, Yuwei, et al. Texture and shape biased two-stream networks for clothing classification and attribute recognition. Proceedings of the IEEE/CVF Conference on Computer Vision and Pattern Recognition. 2020.
> [12] Zhong, Yunshan, et al. Fine-grained data distribution alignment for post-training quantization. European Conference on Computer Vision. Cham: Springer Nature Switzerland, 2022.
> [13] Xu, Shoukai, et al. Generative low-bitwidth data free quantization. Computer Vision–ECCV 2020: 16th European Conference, Glasgow, UK, August 23–28, 2020, Proceedings, Part XII 16. Springer International Publishing, 2020.
> [14] Zhong, Yunshan, et al. Intraq: Learning synthetic images with intra-class heterogeneity for zero-shot network quantization. Proceedings of the IEEE/CVF Conference on Computer Vision and Pattern Recognition. 2022.
> ```

---

> > ### Comment · Reviewer_F97y · 2023-08-10
> > **Thanks for the replies from authors, a follow-up**
> >
> > I have gone through all the replies from the authors, and some of my concerns and questions are well-addressed (Q4, Q5).
> >
> > A1 follow-up: In rebuttal, every author can submit a pdf file containing the images, but I did not see it from the author's rebuttal. It would be good if the visualization in A1 is provided.
> >
> > A2 follow-up: I quickly check the paper [12-14], and there are not too many weighting coefficients to balance different terms. I still encourage the author to conduct an ablation study or sensitivity analysis on these parameters (especially $\alpha_{1,..,5}$)
> >
> > A3 follow-up: For the "texture" in NLP, a concept somewhat analogous to texture in vision could be the distribution of linguistic elements within a text, such as the arrangement of words, phrases, and syntactic structures. This arrangement can provide insights into the text's style. I also agree that trainable filters are a good direction for future work and can help interpretability.

---

> > > ### Author Response · Authors · 2023-08-15
> > > **Response to follow-up A1, A2**
> > >
> > > ### We have sent you the anonymous visualization material via official comment to ACs **(see the top of the page)**, containing Figure A (Visualization of features extracted with LAWS filters) and Figure B (Influence of the trade-off parameters).
> > >
> > > **Visualization of the LAWS texture features**
> > >
> > > Figure A shows the feature map extracted by LAWS texture feature filters.
> > >
> > > * It can be observed that different types of images contain different dominant texture features. For example, crack mainly contains edge features; cloth mainly contains high-frequency points; while the trees with lots of leaves including V-shape, high-frequency points and edge features.
> > >
> > > * We will supplement Figure A in the camera ready version, after introducing the concept of "LAWS texture feature energy" (line 127).
> > >
> > > **Sensitivity analysis on trade-off parameters**
> > >
> > > We display the influence of trade-off parameters in Figure B. For experiment, we follow the settings of Section 4.3 Ablation study (Hyperparameters).
> > >
> > > * In “Step1: Calibration set generation”，The $α_1$ and $α_2$ from Eq. 10 balance different losses in the optimization of calibration samples. We have done a preliminary search for tradeoff parameters when designing this loss. In grid search, we can see that the optimal configurations of these 2 parameters are $α_1$ = 2, $α_2$ = 10.
> > >
> > > * In “Step2: Synthetic samples generation”, the $α_3$, $α_4$, and $α_5$ from Eq. 12 balances the losses in updating the generator for synthetic samples. In grid search, we can see that the optimal configurations of these three parameters are $α_3$ = 0.4, $α_4$ = 0.02 and $α_5$ = 1.8. The configuration is proportional to the previous study FDDA (Fig. 4 in [12], $α_2$ = 0.2 $α_4$ = 0.01 $α_3$ = 0.9). That is, grid search results are consistent under similar frameworks.
> > >
> > > |     $α_1$    |     0    |     1    |     2    |     3    |     4    |
> > > |---|---|---|---|---|---|
> > > |     Acc. (%)    |     42.28    |     48.93    |     50.68    |     49.94    |     49.66    |
> > >
> > > |     $α_2$    |     0    |     5    |     10    |     15    |     20    |
> > > |---|---|---|---|---|---|
> > > |     Acc. (%)    |     43.52    |     47.57    |     50.68    |     49.60    |     49.50    |
> > >
> > > |     $α_3$    |     0    |     0.2    |     0.4    |     0.6    |     0.8    |
> > > |---|---|---|---|---|---|
> > > |     Acc. (%)    |     48.38    |     50.02    |     50.68    |     49.86    |     49.90    |
> > >
> > > |     $α_4$    |     0    |     0.02    |     0.04    |     0.06    |     0.08    |
> > > |---|---|---|---|---|---|
> > > |     Acc. (%)    |     45.73    |     50.68    |     50.41    |     50.01    |     49.35    |
> > >
> > > |     $α_5$    |     0    |     0.6    |     1.2    |     1.8    |     2.4    |
> > > |---|---|---|---|---|---|
> > > |     Acc. (%)    |     47.12    |     48.98    |     49.85    |     50.68    |     49.76    |
> > >
> > > ***
> > >
> > > ```
> > > [12] Zhong, Yunshan, et al. Fine-grained data distribution alignment for post-training quantization. European Conference on Computer Vision. Cham: Springer Nature Switzerland, 2022.
> > > ```

---

### Official Review · Reviewer_yCFw · 2023-07-04

**Soundness:** 3 good
**Presentation:** 3 good
**Contribution:** 3 good
**Rating:** 6
**Confidence:** 4

**Summary:**

The paper points out that there is a strong dependency between the performance of CNN and the texture feature of the dataset. For extending this concept to the quantization field, the paper adopts calibration samples which are trained with manually designed texture filters. In addition to synthetic samples which are generally used in ZSQ works (generated by a network that is trained with Batch Normalization layers’ statistics of a full precision model), the paper exploits calibration samples to quantize a model without the original dataset. Furthermore, the paper applies mixup data augmentation to improve the quantized model’s performance.

**Strengths:**

- This paper is well-motivated and easy to follow.

- The paper adopts a concept of texture feature in Zero-shot quantization for the first time.

- The paper demonstrates the proposed method well with several formulas and figures.

**Weaknesses:**

- It seems costly that generate calibration samples along with synthetic samples for quantizing a neural network.

typo:
- In the 215th row, presnted -> presented

**Questions:**

- In equation 1, only a rounding operation is applied to obtain quantized integers without a clip operation?

- The paper design texture filter manually. Is it possible to get texture filters with the training process?

- Calibration samples are obtained without a generator. Is it possible to generate calibration samples with another generator, or the synthetic sample generator by giving the texture feature energy distribution calibration loss?

- How expensive it is to generate both samples, compared to other works that generate synthetic samples only?

- Qimera [1] executed several experiments with mixup and cutmix, similar with 'Mixup knowledge distillation module' described in 3.2.3, claiming that superposed latent embedding works better than mixup and cutmix. Can a comparison with those methods be provided?


[1] Choi et al. Qimera: Data-free Quantization with Synthetic Boundary Supporting Samples. In Conference on Neural Information Processing Systems. 2021

**Limitations:**

Previous ZSQ works which exploit Batch Normalization layers' statistics for generating synthetic samples are hard to apply transformer-based models.

It is worth analyzing that the proposed method can be applied to those models.

---

> ### Author Rebuttal · Authors · 2023-08-09
>
> ### **We thank the reviewer for the helpful reviews that will help strengthen our paper. Our replies are as follows:**
>
> ---
>
> **Q1: In equation 1, only a rounding operation is applied to obtain quantized integers without a clip operation?**
>
> **A1: Thank you for reminding us that we omitted a clip operation in Eq. 1, and we have revised it.**
>
> ---
>
> **Q2: The paper design texture filter manually. Is it possible to get texture filters with the training process?**
>
> **A2: This is entirely a promising point.**
>
> * With the manually designed texture feature filters, we are the first to introduce the idea of texture calibration and realize it in quantization for better performance.
> * The iterable filters take advantage of automatic design features to facilitate the adaptation of different datasets and tasks. They will be explored in subsequent work and hopefully improve performance.
>
> ---
>
> **Q3: Calibration samples are obtained without a generator. Is it possible to generate calibration samples with another generator, or the synthetic sample generator by giving the texture feature energy distribution calibration loss?**
>
> **A3: These are two possible options. Considering the time cost, we choose to directly extract calibration samples from the model.**
>
> * For introducing another generator, training the generator increases the time and information transfer cost.
>
> * For applying calibration lost to the generator, results in slow iterations and homogeneous samples. So, these two options were not adopted.
>
> ---
>
> **Q4: How expensive it is to generate both samples, compared to other works that generate synthetic samples only?**
>
> **A4: Generating calibration samples requires additional time, which is acceptable for offline PTQ. We generate calibration set (one calibration sample per class) on an NVIDIA GeForce RTX 3090 GPU.**
>
> * For images of size 3×32×32, the generation speed is about 10 seconds per image. For images of size 3×224×224, the generation speed is about 20 seconds per image.
>
> * Taking CIFAR10 as an example, generating the whole calibration set takes 10*10 seconds=100 seconds=1.67 minutes. Parallel processing to accelerate this process is possible.
>
> ---
>
> **Q5: Qimera [11] executed several experiments with mixup and cutmix, similar with 'Mixup knowledge distillation module' described in 3.2.3, claiming that superposed latent embedding works better than mixup and cutmix. Can a comparison with those methods be provided?**
>
> [11] Choi et al. Qimera: Data-free Quantization with Synthetic Boundary Supporting Samples. In Conference on Neural Information Processing Systems. 2021.
>
> **A5: This is a very meaningful question. We will address your concerns with several experiments.**
>
> * First of all, we need to clarify that **the applications of “Mixup knowledge distillation module” and “Superposed latent embeddings/Mixup/Mixcut” are different.** “Superposed latent embeddings/Mixup/Mixcut methods” in Qimera [11] generate **samples with mixed labels to fine-tune the quantization model with cross-entropy loss**. Different from them, we apply the **mixed samples to knowledge distillation with KL divergence loss** of the pre-trained model (teacher model) and the quantized model (student model) (see Eq.13 in Section 3.3.3), **without using mixed labels to fine-tune** the quantized model with cross-entropy.
>
> * Why does superposed latent embedding (Qimera) work better than mixup and cutmix? We observed that traditional mixup and cutmix lead to inaccurate labels. To be detailed, the true label (the inference results of the pre-trained model) of the mixed image is inconsistent with the mixed label. Taking such mixed labels to fine-tune the quantized model with cross-entropy loss tends to be disastrously misleading. A simple experiment as below might be more straightforward. We randomly performed Mixup or Mixcut on 1000 synthetic images and their labels. Similarly, we warm up the generator with 50 epochs and use superposed latent embeddings to generate 1000 synthetic images. Then, we count the consistency of mixed labels and real labels. As shown in Table E, we found that only 34.4% percent of the Mixup or Mixcut samples were correctly labeled, while Qimera's got an advanced Acc. of 66.1%. However, it still can be seen that the labels produced by the above methods are not accurate enough, so we do not recommend using such mixed labels to fine-tune quantization models. Note that we do not use mixed labels to fine-tune the model and do not suffer from such problems.
>
> ---
>
> #### **Tabel E	Comparisons with mix methods on correct label rate (4 bits MobIleNet-V2 on ImageNet)**
> |     Method    |     Correct   label    |     Incorrect   label    |     Correct   rate    |
> |---|---|---|---|
> |     Mixup   or Mixcut    |     344    |     656    |     34.4%    |
> |     Superposed   latent embeddings (Qimera) [11]   |     661    |     339    |     **66.1%   (Best)**    |
>
> ---
>
> * Finally, we would like to compare the effect of superposed latent embeddings/Mixup/Mixcut in our framework. We conduct the comparison on 4 bits MobIleNet-V2 case on ImageNet. As shown in Table F, we find that the above three methods have similar effects. Our proposed “Mixup knowledge distillation module” is slightly ahead of the other two, which takes advantage of decoupling the generator to fuse the entire image, maximizing sample diversity.
>
> ---
>
> #### **Tabel F	Comparisons with mix methods for KL divergence distillation (4 bits MobIleNet-V2 on ImageNet)**
> |     Method    |     Acc.   of quantized model    |     Acc.   of pre-trained model    |     Acc.   loss    |
> |---|---|---|---|
> |     No   augmentation    |     66.21    |     72.49    |     -6.28    |
> |     Mixup   knowledge distillation module (Ours)    |     **67.07  (Best)**    |     72.49    |     -5.42     |
> |     Mixcut                       |     67.01    |     72.49    |     -5.48    |
> |     Superposed   latent embeddings (Qimera)  [11]   |     66.89    |     72.49    |     -5.60    |
>
> ---

---

> > ### Comment · Reviewer_yCFw · 2023-08-17
> >
> > Thank the authors for the answers. I have a question for Table G in the global response. It is obvious that using both kinds of samples maximizes the performance of quantized models. By the way, in Table G, an experiment with calibration samples only shows better results than that with synthetic samples. Because synthetic samples are generated to get a similar distribution to the original dataset, they are likely more helpful for quantization than calibration samples. Can the authors' analysis of the results be provided?

---

> > > ### Author Response · Authors · 2023-08-18
> > > **Reply to Reviewer yCFw**
> > >
> > > **We thank the reviewer for the insightful review of our paper and greatly appreciate the issues raised. Below we provide a detailed analysis of Table G and hope to make it clear.**
> > >
> > > ***
> > >
> > > **Analysis on only synthetic samples case**: Removing the calibration samples means the removal of the calibration method, thus the synthetic sample is uncalibrated.
> > >
> > > **Details:**
> > > * It should be noted that synthetic samples are generated by generator G, who is constrained with ${\mathcal{L}^{G}}$ in Eq. 12 (Section 3.3.2, line 185), where, the $ \mathcal{L}^G_{L-BNS}$ (Eq. 8) and  $\mathcal{L}^G_{D-BNS}$ (Eq. 11) introduce the calibration centers of calibration samples: $\mu_l^C(\hat{x_c}|\hat{y_c}=k)$ and $\sigma_l^C(\hat{x_c}|\hat{y_c}=k)$.
> > >
> > > $$\mathcal{L}^G=\mathcal{L}^G_{CE}+\alpha_3 · \mathcal{L}^G_{BNS}+\alpha_4 ·  \mathcal{L}^G_{L-BNS}+\alpha_{5}  ·  \mathcal{L}^G_{D-BNS} \text{   }  (12)$$
> > >
> > >
> > > * Thus, with calibration samples removed, the $ \mathcal{L}^G_{L-BNS}$ (Eq. 8) and $\mathcal{L}^G_{D-BNS}$ (Eq. 11) will not function. At this point, the synthetic sample loses the calibration information, and thus cannot retain similar texture distribution to the original dataset.
> > >
> > > ***
> > >
> > > **Analysis on only calibration samples case**: The small number of calibration samples causes overfitting.
> > >
> > > **Details:**
> > >
> > > * In Step 1 (line 179), calibration samples capture the preferred texture of each class from full-precision model, which is helpful for quantization. However, with only 1 calibration sample per class, quantized model tends to overfit.
> > > * To address this issue, in Step 2, Generator G generate synthetic samples centered on calibration samples and interfere with Gaussian noise through $\mathcal{L}^G_{D-BNS}$ , thus alleviating the overfit issue and improving the performance.
> > >
> > > To this end, it is obvious that both kinds of samples are essential and with both we can maximizes the performance of quantized models.
> > >
> > > ***
> > >
> > > We would be more than happy to further engage with the reviewer at any time during the discussion period to clear up remaining issues, and also appreciate the reviewer’s willingness to re-evaluate our paper if the concerns are sufficiently addressed.

---

> > > > ### Comment · Reviewer_yCFw · 2023-08-18
> > > >
> > > > Thank the authors for the response. I understand that using both kinds of samples helps enhance the model's performance. However, What I wonder is why the result with calibration samples only shows better performance compared to that of synthetic samples. I know it is hard to be answered. Only the authors' opinions or insights would be OK.

---

> > > > > ### Author Response · Authors · 2023-08-18
> > > > > **Reply to Reviewer yCFw**
> > > > >
> > > > > **Thank you for your feedback, we would like to provide two main reasons to explain why calibration samples work better than synthetic samples:**
> > > > >
> > > > > ***
> > > > >
> > > > > **1. Calibration samples are more accurate with short information link**
> > > > >
> > > > > The calibration samples are directly extracted from the full-precision model F, which reduce the information transfer link of *[ Full-precision model $F$ -> Calibration samples $C$ ]* and become more accurate; while the synthetic samples generated by the generator, with longer information transfer link of *[ Full-precision model $F$ -> Generator $G$ -> Synthetic samples $\bar{D}$ ]*.
> > > > >
> > > > > **2. Calibration samples contains more obvious class information.**
> > > > >
> > > > > The calibration samples are directly optimized with our calibration loss, thus increase the inter-class distance (as shown in Figure 8 (b), black spots denote the calibration samples); while synthetic samples without calibration centers lack enough class information and thus exhibit worse performance.

---

> > > > > > ### Comment · Reviewer_yCFw · 2023-08-19
> > > > > >
> > > > > > Thank the authors for their responses. All my questions and concerns are addressed, and I raise my score to 6.

---

### Official Review · Reviewer_zKQV · 2023-07-06

**Soundness:** 2 fair
**Presentation:** 2 fair
**Contribution:** 2 fair
**Rating:** 5
**Confidence:** 5

**Summary:**

They suggested a zero-shot quantization method to retain the detailed texture feature distribution and introduced the mixup knowledge distillation module to diversify synthetic samples for finetuning

**Strengths:**

They identified the new feature required when generating synthetic data for quantization.

**Weaknesses:**

They should compare their work with MixMix [1], Genie[2] and KW[3].

The authors only empirically showed their superiority. i.e. lt lacks explanations of intuitive or mathematic. The author should give more reasons.

The image they generated showed a little bit of poor quality to argue that it has captured the texture feature distributions. please see the synthetic images in [1], [2], [3].

[1] Li, Yuhang, et al. "Mixmix: All you need for data-free compression are feature and data mixing." Proceedings of the IEEE/CVF International Conference on Computer Vision. 2021.

[2] Jeon, Yongkweon, Chungman Lee, and Ho-young Kim. "Genie: Show Me the Data for Quantization." Proceedings of the IEEE/CVF Conference on Computer Vision and Pattern Recognition. 2023.

[3] Haroush, Matan, et al. "The knowledge within: Methods for data-free model compression." Proceedings of the IEEE/CVF Conference on Computer Vision and Pattern Recognition. 2020.

**Questions:**

1. please provide more detail on why the texture feature distribution calibration is important when generating synthetic data.

2. PTQ methods such as AdaRound, AdaQuant, and Brecq can employ synthetic data to quantize models. how about adapting these post-training quantization schemes for your methods?

3. I expect to see more evaluation on various models in order to convince the superiority of your work.

**Limitations:**

It lacks a literature survey.

---

> ### Author Rebuttal · Authors · 2023-08-09
>
> ### We would like to thank the reviewer for the thoughtful reviews that will help strengthen our paper. In the following, we address each individual question in detail.
>
> **Due to characters limit, all tables for supplementary experiments are displayed in the global response area.**
>
> ***
>
> **Q1: please provide more detail on why the texture feature distribution calibration is important when generating synthetic data.**
>
> **A1: We agreed to provide more details to make it clearer for future readers.**
>
> * For intuitive explanations, we appreciate the comments of reviewer F97y. We will provide straightforward visualization of the texture feature extracted by filters in Figure 5, after introducing the concept of "LAWS texture feature energy" (line 127). In addition, we supplement the actual signal meaning of the filters in Section 3.2.1 (line 126).
>
> * For literature survey and mathematic explanations, we supplement Related work (line 71) on the importance of texture features and the processing details of texture features.
>
> * Many research [4-6] identified the importance of textures feature for CNNs. [4] found that texture representations could capture the statistical characteristics of images for CNNs. [5] showed that classic CNNs were unable to recognize sketches where textures are missing and shapes are left. Similarly, [6-7] validated that CNNs were biased towards textures than shapes, for example, ResNet-50 biased 77.9% texture [6].
>
> * Texture feature extraction is a common method in natural image processing, details can be referred to [8]. Studies [9-10] have shown that image texture features are conductive to image classification and are class separable. We observe that quantized networks suffer from accuracy loss on image that lose texture features (Figure 2). Further, we introduced quantitative indicators of LBP and LAWS texture feature energy to visually demonstrate the texture feature gap between the synthetic and real image (Figure 1,5). Results show that the plug of this gap is beneficial to quantization.
>
> **Q2 and weakness:
> They should compare their work with MixMix [1], Genie[2] and KW[3]. PTQ methods such as AdaRound, AdaQuant, and Brecq can employ synthetic data to quantize models. how about adapting these post-training quantization schemes for your methods?**
>
> **A2: We have also noted the studies[1-3]. We provide comparisons with MixMix [1], Genie[2] and KW[3] below. However, it should be noted that MixMix [1], KW [3] has looser quantization settings and no open-source code, Genie [2] is not the same type of work as ours on feature extraction, which makes the comparison not fair.**
>
> * As for MixMix [1], it focuses on the generalization of synthetic dataset, and **3 pre-trained models** were used for distilling (Section 5, [1]). However, General **ZSQs allow only 1 pre-trained model** to extract data and quantize itself. For KW [3], they quantized the **first & final layers in 8 bits and 1x1 convolution layers in 8 bits** (Table 1-3, [3]), However, most of works including ours quantize **all layers to the same target bits**. Even under unfair comparison, our method exhibits superior performance. As shown in Table A, **we still lead MixMix with a top1 acc. of 3.06%, and lead KW with acc. loss of 0.39% in the 4 bits MobIleNet-V2 case on ImageNet.**
>
> * Genie [2] is a different type of work. They focus on new quantizer (Section 3.2, [2]) and introduced PTQ (AdaRound and BRECQ) for fine-tuning. However, we focus on verifying the idea of texture calibration, regardless of improving the quantizer even though we may achieve better accuracy by optimizing it. Thus, a naive asymmetric quantizer is adopted to maintain consistency with the same type of works. To reach a fair comparison, we adopted 1000 synthetic samples of Genie [2] and conducted the experiment on the same asymmetric quantizer. Results in Table B show that Genie performs similar results to the contemporaneous work. **We lead Genie with 1.79% top1 acc. on the same quantizer in the 4 bits MobIleNet-V2 case.** One of the factors is that Genie ignored class information but we retain class features with dynamic texture calibration.
>
> * On the issue of image visualization, MixMix [1], Genie[2] and KW[3] generate images that conform to human vision. However, our goal is not to generate beautiful images, and we even drop the scale limit of [-1, 1] of the image tensor to expand the feature representation space. Some methods as L2 norm, clamp/clip to generate smooth and visual images were tried but seem no benefit in quantization. Relevant experimental results are shown in Table C.
>
> **Q3: I expect to see more evaluation on various models in order to convince the superiority of your work.**
>
> **A3: We are providing our results for ResNet50 on ImageNet in Table D.**
>
> * **We continue to show advanced performance on the ResNet50, such as we lead AdaDFQ with 7.64%/2.34% Top1 acc. on 3/4 bits case**. We will supplement this result in Table 2.
>
> * For our experiments, we follow the settings of Section 4.1 except that we raise the weight of L-BNS loss and BNS alignment loss to ensure the convergence of the model.
>
> ```
> [1] Mixmix: All you need for data-free compression are feature and data mixing. ICCV. 2021.
> [2] Genie: Show Me the Data for Quantization. CVPR. 2023.
> [3] The knowledge within: Methods for data-free model compression. CVPR. 2020.
> [4] Texture synthesis using convolutional neural networks. NeurIPS. 2015.
> [5] On the performance of GoogLeNet and AlexNet applied to sketches. AAAI. 2016.
> [6] ImageNet-trained CNNs are biased towards texture; increasing shape bias improves accuracy and robustness. arXiv, 2018.
> [7] BiasBed-Rigorous Texture Bias Evaluation. CVPR. 2023.
> [8] Filtering for texture classification: A comparative study. TPAMI. 1999.
> [9] Image Classification Using Laws' Texture Energy Measures. 1987.
> [10] Texture and shape biased two-stream networks for clothing classification and attribute recognition. CVPR. 2020.
> ```

---

> > ### Comment · Reviewer_zKQV · 2023-08-11
> > **Thanks for the detailed response.**
> >
> > Thank you for the response. However, my main concerns still stand as follow:
> >
> > zero-shot "quantization" eventually has to pursue higher accuracy when quantizing models. In this regard, many literatures, related to zero-shot quantization including your work, use and rely on an outdated quantizer used in GDFQ, which is a point that is not acceptable to me.
> > when given only the pre-trained models, practically, users would prefer the quantizer showing better accuracy in a relatively short time.
> > Thus, authors need to prove their scheme on the latest quantizers such as AdaRound, BrecQ, and QDrop (Post-training quantization schemes).
> > According to the paper (Genie), ZeorQ, one of the early works, also showed very good performance when combined with such a PTQ scheme, which would be an example showing PTQ is more suitable for ZSQ.
> > Since MixMix and Genis compared various approaches using BrecQ (which is open-sourced), I would like to encourage the authors to prove the performance of their works by using BrecQ as a quantizer.

---

> > > ### Author Response · Authors · 2023-08-15
> > > **Reply to Reviewer zKQV**
> > >
> > > **We thank the reviewer for the time and effort in our paper and greatly appreciate the issues raised.**
> > >
> > > We will address the reviewer’s concerns below and prove the performance by adopting BrecQ as a quantizer.
> > >
> > > * Table H shows the results for ResNet18/ResNet50/MobileNetV2 on ImageNet. We continue to show advanced performance on these models. The other models in GENIE did not open source their configurations on BrecQ and cannot be re-implement.
> > >
> > > * To this end, our method is proved to show advanced performance in both ZSQ and PTQ. This again verifies the proposed texture calibration method.
> > >
> > > ***
> > >
> > > **Tabel H	 Top-1 acc. (%) of ZSQ methods with BRECQ quantizer (Single model and W4A4 case on ImageNet)**
> > > |     Method    |     MobileNetV2    |     ResNet-18    |     ResNet-50    |
> > > |---|---|---|---|
> > > |     Full   Precision (W32A32)    |     72.49    |     71.08    |     77.00    |
> > > |     ZeroQ+BRECQ    |     49.83    |     69.32    |     73.73    |
> > > |     KW+BRECQ    |     59.81    |     69.08    |     74.05    |
> > > |     IntraQ+BRECQ    |     63.78    |     68.77    |     68.16    |
> > > |     Qimera+BRECQ    |     58.33    |     67.86    |     72.09    |
> > > |     GENIE-D+BRECQ    |     64.68    |     69.70    |     74.89    |
> > > |     **TexQ+BRECQ**    |     **64.94**    |     **69.84**    |     **74.96**    |
> > >
> > > However, it should be noted that our method is not optimized for PTQ fine-tuning, and we focus on synthesizing samples with class texture calibration for ZSQ distillation.
> > >
> > > We would like to thank the reviewer again for the valuable feedback. We will be doing our best to continue to answer any outstanding issues during the discussion period.

---

> > > > ### Comment · Reviewer_zKQV · 2023-08-15
> > > > **Thank you for your effort...**
> > > >
> > > > Thank you for your effort to address my concern. I have some questions or suggestions as below:
> > > >
> > > > 1. could you show the result when bit-width W2A4, where the performance of synthetic data might be more distinguishable?
> > > >
> > > > 2. I would like you to release the source code for someone who does a follow-up study and reproduction even after the final decision of the paper.

---

> > > > > ### Author Response · Authors · 2023-08-17
> > > > > **Reply to Reviewer zKQV (1/2)**
> > > > >
> > > > > # Thank the reviewer for the suggestions. Unfortunately, we believe there still has some misunderstandings about ZSQ/PTQ, that we'd like to sort out before formulating a more detailed response.
> > > > >
> > > > > ***
> > > > >
> > > > > ## 1. Tasks and research ideas
> > > > >
> > > > > ### Group 1: Post-training quantization (PTQ)
> > > > >
> > > > > > PTQ takes calibration data to reduce the error of the model output before and after quantization in layer-wise or block-wise, called PTQ finetuning. To ensure the generalization of the fine-tuned model, **data-free PTQ synthesize calibration images with statistic information.** Relevant research ideas are as follows:
> > > > >
> > > > > * GENIE: Introduce a generator with batch normalization statistics (BNS) alignment loss to synthesize data.
> > > > > * ZeroQ: Introduce BNS alignment loss to optimize Gaussian initialized images.
> > > > >
> > > > > Highlight: statistic information
> > > > >
> > > > > Drawback: low sample diversity, lack class information
> > > > >
> > > > > ### Group 2: Zero-shot quantization (ZSQ)
> > > > >
> > > > > > ZSQ takes calibration data to distill the logits of teacher (pre-trained) and student (quantized) networks. To ensure the complete knowledge transfer, **ZSQ focus on the diversity of synthetic calibration images**, especially class information. Relevant research ideas are as follows:
> > > > >
> > > > > * IntraQ: Introduce cross entropy loss (CE loss) to maintain class information. Increase intra-class heterogeneity of synthetic images with marginal distance constraint.
> > > > > * Qimera: CE loss. Synthesize class boundary supporting samples with superposed latent embeddings.
> > > > > * TexQ: CE loss, Synthesize samples with class texture calibration and Mixup knowledge distillation module.
> > > > >
> > > > > Highlight: sample diversity, class information
> > > > >
> > > > > Drawback: small batch deviation from model statistics
> > > > >
> > > > > **To this end, we have clarified the difference and quite opposite optimization goal for two types of works, group1 (PTQ) and group 2 (ZSQ). These are two different research ideas, making the synthetic samples they produce ```incompatible``` in application.**
> > > > >
> > > > >
> > > > > ***
> > > > >
> > > > > ## 2. Examples of the ```Incompatibility```
> > > > >
> > > > > ### Example1: W4A4 case
> > > > >
> > > > > We would like to give some examples that apply methods to mismatching task as in Table I, to clarify these two **opposite optimization goals leads to ```incompatible```**.
> > > > >
> > > > > * Taking Group 1 for ZSQ distillation task: As shown in Table I, for the advanced GENIE-D and ZeroQ in group1, the homogenized sample achieves lagging performance in the ZSQ distillation task.
> > > > > * Taking Group 2 for PTQ fine-tuning task: The advanced Qimera in group2 produces diversified class boundary supporting samples and achieves good performance in ZSQ, however, they get the worst results in PTQ task in Table I.
> > > > >
> > > > > **```This phenomenon is caused by the different optimization goals of the two groups, making them two types of works.```**
> > > > >
> > > > > ***
> > > > >
> > > > >
> > > > > **Tabel I	Top-1 acc. (%) of ResNet18 (W4A4 case on ImageNet)**
> > > > > |     Group    |     Method    |     Class   information    |     Highlight    |     PTQ (BRECQ)       |     ZSQ   (Distill)    |
> > > > > |---|---|---|---|---|---|
> > > > > |          |     Full   Precision    |          |     -    |     71.08    |     71.47    |
> > > > > |     ---    |     ---    |     ---    |     ---    |     ---    |     ---    |
> > > > > |     1    |     GENIE-D      |     x    |     BNS   loss and swing convolution    |     69.70   $^‡$    |     ```59.96```   $^§$    |
> > > > > |     1    |     ZeroQ    |     x    |     Gaussian   initialization and BNS loss    |     69.23   $^‡$    |     ```22.58```   [Q]    |
> > > > > |     ---    |     ---    |     ---    |     ---    |     ---    |     ---    |
> > > > > |     2    |     Qimera       |     √    |     Class   boundary supporting samples    |     ```67.86```$^‡$    |      63.84 [Q]    |
> > > > > |     2    |     IntraQ       |     √    |     Synthetic   images with intra-class heterogeneity    |     ```68.77```$^‡$    |     66.47   [I]    |
> > > > > |     ---    |     ---    |     ---    |     ---    |     ---    |     ---    |
> > > > > |     2    |     TexQ (Ours)    |     √    |     Class   texture calibration and Mixup knowledge distillation    |     69.84    |     67.73    |
> > > > >
> > > > > *```Results```: Results on incompatible/mismatching task. For example, take group1's method for ZSQ, or take group2's method for PTQ.*
> > > > >
> > > > > *‡ Taken from [G]*
> > > > >
> > > > > *§ Reproduced under the same ZSQ quantization framework*
> > > > >
> > > > > ```
> > > > > [G] Genie: Show Me the Data for Quantization. CVPR. 2023.
> > > > > [I] Intraq: Learning synthetic images with intra-class heterogeneity for zero-shot network quantization. CVPR. 2022.
> > > > > [Q] Qimera: Data-free Quantization with Synthetic Boundary Supporting Samples. NeurIPS. 2021.
> > > > > ```
> > > > >
> > > > > ***

---

> > > > > ### Author Response · Authors · 2023-08-17
> > > > > **Reply to Reviewer zKQV (2/2)**
> > > > >
> > > > > ## 2. Examples of the ```Incompatibility``` Part 2
> > > > >
> > > > > ### Example2: W2A4 case
> > > > > The mismatching goal between the two tasks amplify their incompatibility in W2A4 case.
> > > > > In Table J, Qimera and IntraQ focus on diversified sample, which is very conducive to ZSQ distillation method but not for PTQ, and therefore suffer from the serious incompatible problem.
> > > > >
> > > > > ***
> > > > >
> > > > > **Tabel J	Top-1 acc. (%) of 2 groups of methods on PTQ task  (W2A4 case on ImageNet)**
> > > > > |     Group    |     Method    |     Class   information         |     ResNet-18    |
> > > > > |---|---|---|---|
> > > > > |          |     Full   Precision (W32A32)    |     -    |     71.08    |
> > > > > |     1    |     GENIE-D+BRECQ    |     x    |     64.24    |
> > > > > |     1    |     ZeroQ+BRECQ    |     x    |     61.63    |
> > > > > |     ---    |     ---    |     ---    |     ---    |
> > > > > |     2    |     IntraQ+BRECQ    |     √    |     53.39    |
> > > > > |     2    |     Qimera+BRECQ    |     √    |     47.80    |
> > > > >
> > > > > ***
> > > > >
> > > > > ## 3. Our results in W2A4 case
> > > > >
> > > > > We provide our results on BRECQ in W2A4 case, which also support the view on incompatibility. As in Table K, similar results among ZSQ methods are displayed.
> > > > >
> > > > > ***
> > > > >
> > > > > **Tabel K	Top-1 acc. (%) of ZSQ methods on incompatible PTQ (BRECQ) task (W2A4 case on ImageNet)**
> > > > > |     ZSQ method    |     MobileNetV2    |     ResNet-18    |     ResNet-50    |
> > > > > |---|---|---|---|
> > > > > |     Full   Precision (W32A32)    |     72.49    |     71.08    |     77.00    |
> > > > > |     IntraQ    |     35.38    |     55.39    |     44.78    |
> > > > > |     Qimera    |     3.73    |     47.80    |     49.13    |
> > > > > |     TexQ    |     22.79    |     55.70    |     51.33    |
> > > > >
> > > > > ***
> > > > >
> > > > > ## 4. Our effort
> > > > >
> > > > > Finally, there are still many research gaps in distillation-based ZSQ, such as distillation mode and image feature extraction, which need researchers to further explore and improve the accuracy. Our efforts aim at providing insights in texture feature extraction for ZSQ research community. We plan to release the our code after accepted.
> > > > >
> > > > > ***
> > > > >
> > > > > ### Thank you again for your feedback. We strongly hope that our reply has satisfied your need for clarification and alleviated your doubts.

---

### Author Rebuttal · Authors · 2023-08-09

## **[Global Response] Tables for supplementary experiments**

************************************************

### **Tabel A	Comparisons with MixMix and KW (4 bits MobIleNet-V2 on ImageNet)**
|     Method    |     Settings    |     Acc.   of quantized model    |     Acc.   of pre-trained model    |     Acc.   loss    |
|---|---|---|---|---|
|     Ours    |     1   model, all layers in 4 bits    |     **67.07  (Best)**    |     72.49    |     -5.42     |
|     MixMix   [1]    |     3   model, all layers in 4 bits    |     64.01   [1]    |     72.49       |     -8.48    |
|     KW   [3]    |     1   model, first & final layers in 8 bits    |     66.07   [3]    |     71.88       |     -5.81    |

[1] Mixmix: All you need for data-free compression are feature and data mixing. ICCV. 2021.

[3] The knowledge within: Methods for data-free model compression. CVPR. 2020.

---

### **Tabel B	Comparison with Genie on the same quantizer (4 bits MobIleNet-V2 on ImageNet)**
|     Synthetic   data    |     Acc.   of quantized model    |     Acc.   of pre-trained model    |     Acc.   loss    |
|---|---|---|---|
|     Ours    |     **67.07  (Best)**     |     72.49    |     -5.42   |
|     Genie   (CVPR 2023)  [2]    |     65.28    |     72.49    |     -7.21    |
|     AdaDFQ   (CVPR 2023)    |     65.41    |     72.49    |     -7.08    |

[2] Genie: Show Me the Data for Quantization. CVPR. 2023.

---

### **Tabel C	Comparisons on different synthetic image constraints (4 bits MobIleNet-V2 on ImageNet)**
|     Method    |     Settings    |     Acc.   of quantized model    |     Acc.   of pre-trained model    |     Acc.   loss    |
|---|---|---|---|---|
|     Our    |     Texture   calibration, Unrestricted tensor range    |     **67.07  (Best)**     |     72.49    |     -5.42    |
|     Our+clamp    |     Texture   calibration, Restricted tensor range of [-1, 1]    |     66.26    |     72.49    |     -6.23    |
|     Genie   [2]    |     Restricted   tensor range of [-1, 1]    |     65.28    |     72.49    |     -7.21    |

[2] Genie: Show Me the Data for Quantization. CVPR. 2023.

---

### **Table D. Top 1 accuracy (%) results of ResNet50 on ImageNet.**
|     Methods    |     W4A4    |     W3A3    |
|---|---|---|
|     GDFQ(ECCV   2020)    |     54.16    |     0.31    |
|     ZAQ   (CVPR 2021)    |     53.02    |     -    |
|     ARC   (IJCAI 2021)    |     64.37    |     1.63    |
|     Qimera   (NeurIPS 2021)    |     66.25    |     -    |
|     ARC+AIT   (CVPR 2022)    |     68.27    |     -    |
|     AdaSG   (AAAI 2023)    |     68.58    |     16.98    |
|     AdaDFQ   (CVPR 2023)    |     68.38    |     17.63    |
|     **TexQ   (Ours)**    |     **70.72**    |     **25.27**    |

---

### **Tabel E	Comparisons with mix methods on correct label rate (4 bits MobIleNet-V2 on ImageNet)**
|     Method    |     Correct   label    |     Incorrect   label    |     Correct   rate    |
|---|---|---|---|
|     Mixup   or Mixcut    |     344    |     656    |     34.4%    |
|     Superposed   latent embeddings (Qimera) [11]   |     661    |     339    |     **66.1%   (Best)**    |

[11] Qimera: Data-free Quantization with Synthetic Boundary Supporting Samples. NeurIPS. 2021.

---

### **Tabel F	Comparisons with mix methods on TexQ (4 bits MobIleNet-V2 on ImageNet)**
|     Method    |     Acc.   of quantized model    |     Acc.   of pre-trained model    |     Acc.   loss    |
|---|---|---|---|
|     No   augmentation    |     66.21    |     72.49    |     -6.28    |
|     Mixup   knowledge distillation module (Ours)    |     **67.07  (Best)**    |     72.49    |     -5.42     |
|     Mixcut                      |     67.01    |     72.49    |     -5.48    |
|     Superposed   latent embeddings (Qimera)  [11]   |     66.89    |     72.49    |     -5.60    |

[11] Qimera: Data-free Quantization with Synthetic Boundary Supporting Samples. NeurIPS. 2021.

---

### **Tabel G	Three possible sample synthesis schemes for TexQ (4 bits MobIleNet-V2 on ImageNet)**
|     Method    |     Acc.   of quantized model    |     Acc.   of pre-trained model    |     Acc.   loss    |
|---|---|---|---|
|     Calibration   samples + Synthetic samples    |     **67.07  (Best)**    |     72.49    |     -5.42     |
|     Only   synthetic samples    |     65.42    |     72.49    |     -7.07    |
|     Only   calibration samples    |     66.04    |     72.49    |     -6.45    |

---

### Author Response · Authors · 2023-08-15
**Anonymous visualization material for Reviewer F97y**

Dear Chairs,

Reviewer F97y encourage us to submit a visualization file, and we would like to provide him with the anonymous PDF link below:
https://anonymous.4open.science/r/3309-F2AF/3309_rebuttal_20230815.pdf

Thank you!

---

### Decision · Program_Chairs · 2023-09-21

**Decision:**

Accept (poster)

**Comment:**

The paper titled "TexQ: Zero-shot Network Quantization with Texture Feature Distribution Calibration" proposes a novel approach to zero-shot quantization (ZSQ) by addressing the challenge of retaining detailed texture feature distributions in synthetic samples. The authors introduce the concept of texture feature energy distribution calibration to guide the synthesis of calibration images, and they employ a mixup knowledge distillation module to enhance the diversity of synthetic samples for fine-tuning. The paper showcases strong performance in ultra-low bit width quantization on benchmark datasets such as CIFAR10/100 and ImageNet.

Reviewer zKQV acknowledges the paper's contributions and highlights its identification of the need for new features when generating synthetic data for quantization. The strengths include the novel approach to texture feature distribution calibration and the use of mixup for knowledge distillation. However, the reviewer raises concerns regarding the lack of comparisons with related works, as well as the empirical nature of the paper without extensive intuitive or mathematical explanations. The reviewer also comments on the quality of generated images and requests more evaluation on various models to demonstrate the superiority of the proposed method. In their rebuttal, the authors address each concern raised by the reviewer and other reviewers, providing detailed explanations and additional experimental results. They emphasize the distinction between post-training quantization (PTQ) and zero-shot quantization (ZSQ) methods, highlighting the different optimization goals and resulting incompatibility in sample synthesis. The authors compare their approach with MixMix, Genie, and KW, discussing how the unique nature of their proposed method affects these comparisons. They provide results on various models and quantizers to demonstrate their method's performance. The reviewer acknowledges the authors' efforts in addressing concerns and understanding the distinction between PTQ and ZSQ. While maintaining their concerns about outdated quantizers, the reviewer raises their rating and recognizes the progress and importance of the proposed texture feature calibration approach. In the later stages, the reviewer is willing to increase the score to borderline accept but reiterates their earlier concerns about using outdated quantizers for ZSQ.

Reviewer yCFw acknowledges the paper's strong motivation and clear presentation. The introduction of texture features in ZSQ is highlighted as a significant contribution, along with the effective demonstration of the proposed method through experiments. The reviewer raises concerns about the cost of generating both calibration and synthetic samples, prompting the authors to provide insights into the time and computational requirements. The reviewer also suggests comparing the proposed mixup knowledge distillation module with other methods like superposed latent embeddings from Qimera (which was also requested in Reviewer zKQV's questions). In response, the authors differentiate the applications of these techniques and provide a thorough comparison of their proposed method with superposed latent embeddings, Mixup, and Mixcut, addressing label accuracy and performance. The reviewer also inquires about the effectiveness of using only calibration samples compared to synthetic samples. In their rebuttal, the authors provide an insightful analysis, explaining that calibration samples offer greater accuracy due to shorter information transfer links and more obvious class information. This explanation clarifies the observed performance difference between the two types of samples. Finally, the reviewer appreciates the authors' responses, and after receiving clarifications, raises their rating.

Reviewer F97y raises several concerns, including the need for improved clarity in introducing the concept of "LAWS texture feature energy." In their rebuttal, the authors agree to provide more background and visualization to clarify the concept. They also address the reviewer's query about the significance of these texture features and their importance in retaining texture information. The authors provide additional explanations and promise to enhance the discussion around this aspect. The reviewer suggests that the paper could benefit from conducting an ablation study or sensitivity analysis on the weighting coefficients used to balance different loss functions. In response, the authors acknowledge the suggestion and explain their empirical parameter settings. They also justify their focus on verifying the texture feature distribution calibration method and its compatibility with similar studies. Regarding the terminology "data-free quantization (DFQ)" vs. "zero-shot quantization (ZSQ)," the authors confirm their equivalence and agree to clarify these terms in the paper's related work section. In the follow-up comments, the reviewer acknowledges that some concerns have been well-addressed, particularly the clarification of terminology and the inclusion of insights on transferring the method to other modalities. They also raise specific points related to visualization and parameter analysis. The authors respond by providing visualization material for the LAWS texture features and demonstrating the influence of trade-off parameters through sensitivity analysis (Though ACs are not certain that sharing the material with the official comment to ACs channel is the right way to do this). Overall, the reviewer finds that some of their concerns have been effectively addressed and acknowledges the authors' responses.

Reviewer qhGS suggests considering more widely used architectures, such as ResNet-50, in experiments, and expanding validation to other tasks like object detection on datasets like COCO. Additionally, the reviewer raises questions regarding the generalization of the proposed method and its applicability to Transformer structures. In the authors' rebuttal, they include ResNet-50 in the revised version, showcasing competitive results. Regarding the Transformer structure, they indicate plans for introducing quantization for it in future work. The reviewer increases their score to 7, expressing support for accepting the paper.

Finally, reviewer h8LC points out specific confusing statements, inconsistencies, and unclear explanations in the text. They also raise questions about the role of the calibration center in synthetic sample generation and the need for synthetic samples when a calibration set is available. Additionally, they inquire about the distribution of texture elements in synthetic images and the fairness of comparison between synthetic data from the generator and direct optimization. In their rebuttal, the authors address each point raised by the reviewer and provide explanations and revisions to enhance the paper's clarity. In the final comment, reviewer expresses willingness to raise their rating.

In general, the authors' rebuttal was effective in persuading the reviewers to support acceptance. The ACs align with the reviewers' assessments and recommend acceptance.